# Systematic analysis of snRNA genes reveals frequent *RNU2-2* variants in dominant and recessive developmental and epileptic encephalopathies

Small nuclear RNAs (snRNAs) are essential components of the spliceosome. De novo variants in snRNA genes *RNU4-2* (ReNU syndrome), *RNU5B-1* and *RNU2-2* have been linked to dominant neurodevelopmental disorders (NDDs), revealing a large unexpected contribution of noncoding RNA genes to genetic diseases. Here, through international collaborations, we analyze systematically 200 potentially functional snRNA genes in a French cohort of 34,329 people with rare disorders. We report *RNU2-2* variants in 141 individuals, including 35 with recurrent dominant pathogenic variants and 91 affected members from 73 families with biallelic variants. Recessive *RNU2-2* NDD is at least twice as frequent as the dominant form and often involves a de novo variant in trans with an inherited allele, consistent with the high mutability of snRNA genes. Dominant and recessive *RNU2-2* NDDs share overlapping clinical features, with frequent epilepsy. Blood transcriptomics and DNA methylation analyses revealed subtle, variant-specific effects on splicing and episignatures. Our results support a gradient-of-impact model bridging dominant and recessive inheritance, and establish *RNU2-2* variants as a principal contributor to NDDs, nearly as prevalent as ReNU syndrome.

Small nuclear RNAs (snRNAs) are noncoding RNAs essential for RNA processing and splicing of premessenger RNAs (mRNA). The spliceosome—a dynamic ribonucleoprotein (RNP) complex that catalyzes splicing—depends on five uridine-rich snRNAs for its assembly and function. In mammals, two spliceosome types operate according to intron class: the major spliceosome excises >99% of introns with GU–AG splice sites (U2-type) using snRNAs U1, U2, U4, U5 and U6, whereas the minor spliceosome removes rare introns with AU–AC or GU–AG splice sites (U12-type) with snRNAs U11, U12, U4atac and U6atac, sharing U5 with the major complex[1–3].

Although minor spliceosome snRNA genes are single-copy, major spliceosome genes have several functional copies[4]. Human genomes contain at least two U4, five U5 and five U6 functional genes. U1 is expressed from at least four identical copies (*RNU1-1* to *RNU1-4*) on chromosome (chr) 1p36.13, plus more than 30 variant U1 (*RNVU1*)

genes on 1q21.1–q23.3[5–7]. Most U2 genes ('*RNU2-1*') are organized in large tandem arrays on chr17q21.31, with copy numbers ranging from 6 to >80 per chromosome[8].

Biallelic pathogenic variants were first described in *RNU4ATAC* and cause phenotypically variable developmental disorders: microcephalic osteodysplastic primordial dwarfism type I, Taybi–Linder, Lowry–Wood or Roifman syndromes[9–13]. Recessive variants in *RNU12* may lead to craniosynostosis, anal anomalies and skin lesions and/or spinocerebellar ataxia[14,15] although definitive evidence is missing.

In 2024, de novo variants in the major spliceosome gene *RNU4-2* were shown to cause ReNU syndrome (OMIM 620851)—one of the most common known NDDs[16,17]. These dominant variants are located within an 18-bp critical region spanning the T loop and part of stem III, facing the U6 ACAGAGA box that enables 5' splice site (5'SS) recognition[16,18].

✉e-mail: christel.depienne@uk-essen.de; gaetan.lesca@chu-lyon.fr; caroline.nava@aphp.fr

**Fig. 1 | Systematic in silico analysis of possible functional snRNAs. a,** Filtering strategy used to retain genes expressing possible functional snRNAs. **b,c,** Number and distribution of annotated spliceosomal snRNAs before (**b**) and after (**c**) filtering. **d,** Expression of snRNAs in the human brain from ENCODE small RNA-seq data. Top: all mapped reads (including multimapped). Bottom: uniquely mapped reads. Expression is shown as log$_{10}$ of the maximum normalized RNA-seq signal. Colors indicate the proportion of the snRNA length covered by mapped reads: 100% (purple), 75–99% (dark blue), 50–74% (green), <50% (yellow). Genes in red correspond to annotated pseudogenes. Asterisk, genes expressing U2-1 copies within the chr1 cluster lack HGNC approved symbols and were numbered in ascending order of their genomic coordinates. NA, not applicable.

*RNU4-2* pathogenic variants are associated with mild, but specific, widespread splicing and methylation abnormalities in blood cells of affected people, the degree of which correlates with disease severity[16,18]. De novo variants in *RNU5B-1* (and possibly *RNU5A-1*) clustering in the U5 5′ loop I lead to NDD with variable malformations[18,19]. The recent discoveries of biallelic variants in other regions of *RNU4-2* in NDD patients and heterozygous variants in *RNU4-2* (and genes expressing U6) in families with retinitis pigmentosa[20] has expanded its mutational spectrum and added complexity to variant interpretation[21,22].

Identifying variants in snRNA genes is complicated by sequence redundancy and incomplete annotations, especially when distinguishing functional genes from pseudogenes. De novo pathogenic variants in *RNU2-2P*, initially annotated as a pseudogene, were linked recently to a dominant NDD with epilepsy[19,23]. Because *RNU2-2P* is expressed at levels similar to *RNU2-1*, it was reclassified as *RNU2-2*[19,23,24]. However, unlike *RNU4-2*, no splicing anomalies were detected in blood transcriptomes of patients with *RNU2-2* variants[23]. Strikingly, active snRNA (and tRNA) genes are hypermutable, with up to tenfold more de novo variants than other genomic regions—a characteristic that may help distinguish functional genes from pseudogenes[25].

In this study, we build on our previous work encompassing 50 Human Genome Organization (HUGO) Gene Nomenclature Committee (HGNC)-approved snRNA genes[18], extending it to investigate variants in possibly functional snRNA genes systematically in a large French cohort with rare diseases. This led us to report a highly prevalent recessive NDD with epilepsy caused by biallelic variants in *RNU2-2* that was validated through international collaborations.

## Results

### Identification of potentially functional spliceosomal snRNA genes

To distinguish functional snRNA genes from pseudogenes, we performed an in silico analysis of all annotated snRNA genes (Methods). The Ensembl database contains 2,094 snRNA genes in the hg38 reference genome: 1,741 are spliceosomal snRNA genes, and the remainder have other functions or are not on identifiable chromosomes. We prioritized genes overlapping proximal *cis*-regulatory elements (cCREs; from ENCODE[26]), reported as hypermutable[25], or HGNC-approved[18]. This analysis yielded 200 potentially functional snRNA genes, including 147 'pseudogenes' and *RNU2-2* (Fig. 1a and Supplementary Table 1).

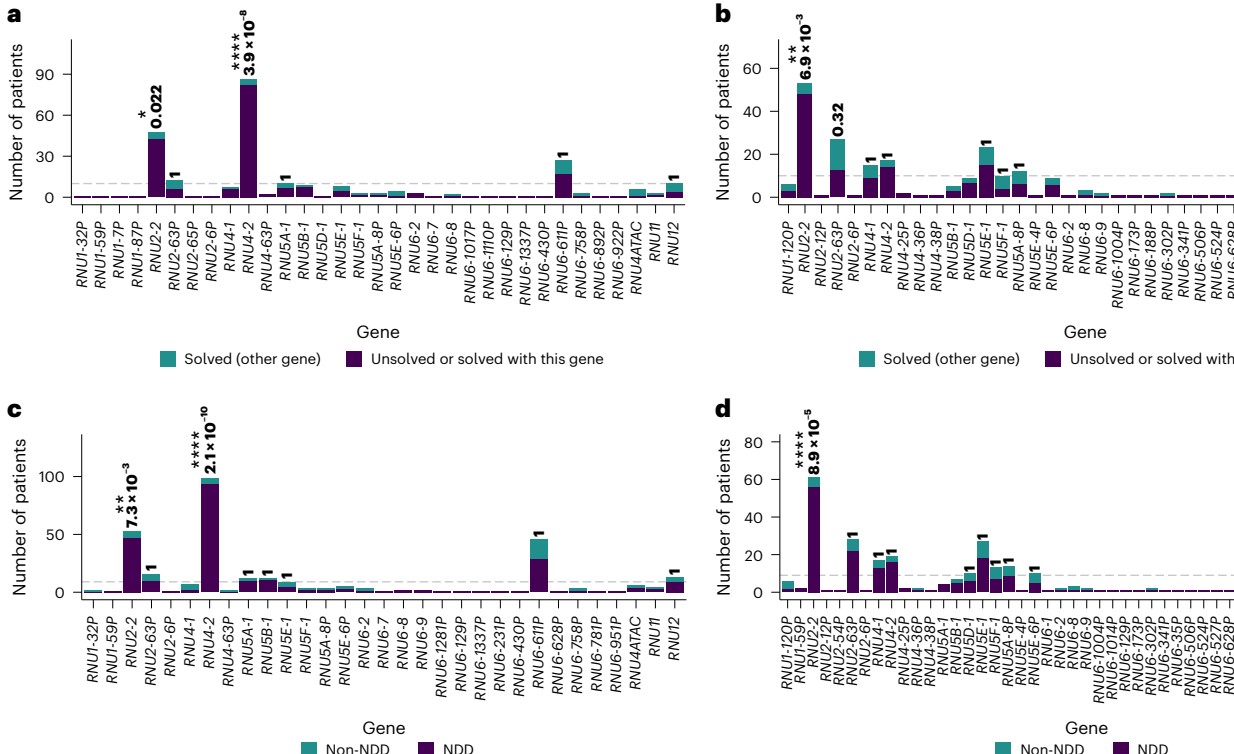

**Fig. 2 | Identification of potential new snRNA gene–disease associations in the PFMG cohort. a,b,** The cohort was divided into solved (*n* = 9,180) and unsolved (*n* = 20,735) cases for discovery analyses, with cases solved by variants in snRNAs with known disease association merged into the unsolved group. We compared the proportion of cases with rare variants (gnomAD allele count <100) between solved and unsolved groups for rare de novo variants (**a**) and rare biallelic variants (**b**). **c,d,** The cohort was divided into NDD (*n* = 22,775) and non-NDD (*n* = 11,554) cases for discovery analyses. We compared the proportion of cases with rare variants (gnomAD allele count <100) between NDD and non-NDD groups for rare de novo (**c**) and rare biallelic (**d**) variants. Two-sided Fisher's tests

were used to test statistical enrichment in unsolved versus solved cases for genes with at least ten patients carrying variants (**a,b**) and in NDD versus non-NDD cases for genes with at least nine such patients (**c,d**) (dashed gray line: minimum number of patients needed to reach statistical significance in the cohort). The number of patients per group and gene are shown in Supplementary Tables 2 and 4. Correction for multiple comparisons using Bonferroni was performed independently for unsolved versus solved cases and NDD versus non-NDD cases. Adjusted *P* values are shown above the bars. *$P$ = 0.01–0.05; **$P$ = 0.001–0.01; ****$P$ < 0.0001.

The breakdown of snRNA types aligns with previous observations: minor spliceosome snRNAs exist mainly as single copies (range 1–3), whereas major spliceosome snRNAs are present in many more copies (range 11–117). U6 has the highest number of copies and U5, shared by both complexes, is intermediate (*n* = 9; Fig. 1b,c).

In parallel, we reanalyzed an ENCODE brain small RNA sequencing (RNA-seq) dataset[27]. We performed two analyses: one using all mapped reads, including multimapped, and another restricted to uniquely mapped reads. The first captures expression from all identical copies, whereas the second reflects only uniquely assignable snRNAs. In total, 87 putatively functional snRNAs, including 39 annotated as pseudogenes, were detectable in the human brain (Fig. 1d), raising the possibility that more NDDs are driven by snRNA variants.

### Analysis of de novo and biallelic variants in putatively functional snRNA genes in the Plan France Médecine Génomique 2025 cohort

We next analyzed variants in the 200 potentially functional snRNA genes in the Plan France Médecine Génomique 2025 (PFMG) cohort[28], which comprised short-read genome data from 34,329 patients with rare disorders (22,775 with NDDs). We focused on de novo and biallelic variants, hypothesizing roles in dominant or recessive monogenic disorders. After accounting for artefacts from short-read mapping and excluding low-quality variants (Supplementary Notes), 843 high-confidence variants (330 de novo, 551 biallelic) in 66 genes were identified in 616 patients (Extended Data Figs. 1–3 and Supplementary Figs. 1–4).

To identify possible new disease gene associations, we divided the cohort into solved (*n* = 9,180) and unsolved (*n* = 20,735) cases. Pathogenic (P) and likely pathogenic (LP) variants in disease-associated snRNA genes (*RNU4ATAC*, *RNU4-2*, *RNU5B-1*, *RNU2-2*) were curated manually and reassigned to the unsolved category, as they would have been before gene–disease association. We assessed both de novo and biallelic variants in the 66 genes focusing on rare variants (allele counts <100 in gnomAD v.3; Extended Data Fig. 3c). De novo variants were enriched significantly in *RNU4-2* and *RNU2-2* when comparing solved and unsolved cases (or gene-solved cases). For biallelic variants, *RNU2-2* showed significant enrichment across combined cohorts (*P* = 6.9 × 10⁻³, two-sided Fisher's test), whereas *RNU4ATAC* reached significance only in the SeqOIA subcohort (Fig. 2a,b, Extended Data Fig. 4, Supplementary Fig. 5 and Supplementary Tables 2 and 3). We then compared NDD (*n* = 22,775) and non-NDD (*n* = 11,554) cases. In the entire PFMG cohort, de novo variants were enriched significantly in *RNU4-2* and *RNU2-2*, whereas biallelic variants were enriched only in *RNU2-2* (*P* = 8.9 × 10⁻⁵, two-sided Fisher's test) (Fig. 2c,d, Extended Data Fig. 4, Supplementary Fig. 6 and Supplementary Tables 4 and 5). These results indicate that *RNU2-2* variants probably contribute to both dominant and recessive NDDs, similar to *RNU4-2*[21,22].

### *RNU2-2* variants as a cause of dominant and recessive disorders
To refine the search for pathogenic variants, we reanalyzed the PFMG cohort restricting only the homozygote frequency (<3 in gnomAD v.3, <5 in internal databases), then applied filters on allele frequencies in All

of Us (AoU) database (AC < 50 for de novo and <200 for biallelic variants) (Supplementary Fig. 7). Using these criteria, 42 unrelated patients had rare de novo *RNU2-2* variants (Supplementary Table 6). Twenty-one unrelated patients and one monozygotic twin harbored previously reported pathogenic alleles: n.4G>A (*n* = 11) and n.35A>G (*n* = 11 including the twin). One patient with n.35A>G was mosaic, and deep-targeted sequencing (>2,000×) confirmed mutant allele fractions of 12% (678 of 5,218 reads) in blood, 20% (1,181 of 5,659 reads) in urine and 25% (574 of 2,267 reads) in buccal cells. In addition, n.4G>A was observed in a singleton. The remaining 21 patients had other de novo variants (Fig. 3 and Supplementary Notes). In 7 of 21 individuals (with n.5C>A, n.6T>C, n.7_8insA, n.31G>A, n.37T>G (*n* = 2) and n.40C>T), further examination revealed a second rare variant that was always in *trans*, suggesting recessive inheritance. In contrast, a single patient with n.4G>A had a second variant in *trans* (n.80A>G; Supplementary Table 6). In addition to these seven cases, 45 probands had rare biallelic variants in the PFMG cohort.

To expand the *RNU2-2* variant spectrum, we reanalyzed available genome data and/or performed targeted sequencing of *RNU2-2* in 5,456 people with NDDs and identified additional cases with *RNU2-2* variants through *seqr*[29] (Supplementary Table 7). This large international collaborative effort revealed 34 additional unrelated patients: 13 had monoallelic variants, including 9 patients with n.4G>A (8 de novo and 1 nonmaternal), 3 n.35A>G variants (2 de novo) and 1 with de novo n.38A>G; 21 families had biallelic variants (Extended Data Fig. 1b).

Combined with the PFMG cohort, we report a total of 141 patients from 122 unrelated families carrying 96 distinct point variants in *RNU2-2* that we classified using American College of Medical Genetics and Genomics criteria (Supplementary Table 8) and two partial-/whole-gene deletions: 35 patients (including the twin) had the dominant n.4G>A and n.35A>G pathogenic variants; 15 patients had another single de novo heterozygous variant; 91 patients from 73 unrelated families exhibited biallelic variants. Of these, 54 carried compound heterozygous variants, 17 had homozygous variants and 2 harbored hemizygous variants associated in trans with a complete or partial gene deletion (Fig. 3 and Extended Data Fig. 5). In 16 families, one or more siblings were also affected, with both variants cosegregating with the disease in affected siblings (Fig. 4a and Extended Data Fig. 6). Twenty-two recessive variants were found in at least two unrelated families, including n.20G>A, n.40C>G, n.40C>T, n.45C>T, n.100T>C, n.107_118del in at least five families and 16 additional variants in two or three families (Supplementary Notes and Supplementary Table 8).

Genome data were available for 110 of the 122 index cases. A single person had a de novo nonsense variant in *GATA3*, partially explaining his phenotype; all others had remained unsolved. Overall, 104 of the 110 patients presented with a NDD phenotype. Of the six remaining cases, three harbored de novo monoallelic variants (one person with cancer, one fetus with a cerebral malformation and one non-NDD patient) and three had biallelic variants (one terminated fetus and two non-NDD patients, including the person with the *GATA3* variant). In the PFMG cohort, NDD was the predominant presentation, observed in 82 of 88 patients ($P = 1.89 \times 10^{-18}$, one-tailed binomial test; 95% confidence interval, 0.87–1.00).

Twenty-six de novo variants could be phased reliably: 16, including all phaseable occurrences of the pathogenic n.4G>A (*n* = 8) and n.35A>G (*n* = 2), along with n.5C>A, n.6T>C and n.7_8insA (*n* = 2), n.31G>A and n.37T>G were phased to the maternal allele. The remaining ten (n.6T>C; n.21C>G; n.32T>G; n.37T>G; n.40C>T, *n* = 3; n.62T>G; n.129_139del; n.143_167del) arose de novo on the paternal allele (Supplementary Table 6).

### Dominant and recessive *RNU2-2* NDDs share overlapping features

We next aimed to delineate the clinical spectrum associated with both dominant and recessive *RNU2-2* disorders. Detailed clinical data were collected for 112 patients (55 female, 57 male), including 20 with n.4G>A, 12 with n.35A>G, and 7 with another monoallelic de novo variant, as well as 73 patients from 55 unrelated families with biallelic inheritance (Table 1 and Supplementary Tables 9–11). The median age at inclusion in the study was 13 years (range: 0 (fetus) to 46 years).

Overall, considering only patients with dominant pathogenic (*n* = 32, including the twin) and biallelic variants (*n* = 73), all patients with available data had developmental delay, and all older than 3 years had intellectual disability (ID) except one with developmental delay without ID, showing fragile visuoconstructive reasoning (n.20G>A/n.145A>G). Severe/profound ID was most frequent (75 of 97, 77%), followed by moderate (17 of 97, 18%) and mild (4 of 97, 4%) ID. Epilepsy occurred in 85% patients (88 of 104), with identical rates in dominant (28 of 32, 88%) and recessive (60 of 72, 83%) cases. Age at seizure onset ranged from 8 weeks to 16 years (median: 1.5 years); 82% (72 of 88) had seizure onset between 8 weeks and 3 years (monoallelic: 22 of 28 biallelic: 50 of 60), while 18% (16 of 88) had seizures after 3 years (monoallelic: 6 of 28; biallelic: 10 of 60). Seizure types were variable. In biallelic families, 39 of 44 (89%) had generalized seizures, whereas in monoallelic cases, 19 of 24 (79%) had focal seizures, including hemicorporal seizures in 7 of 23 (30%). Generalized tonic-clonic seizures occurred in 38 of 60 patients (63%; biallelic 28 of 37, monoallelic 10 of 23), myoclonic seizures in 75% versus 29%, epileptic spasms in 43% versus 25%, and absence seizures in 32 of 63 patients (51%; monoallelic 10 of 24, biallelic 22 of 39). Patients were treated according to their seizure type, as is standard clinical practice, with 60 of 82 patients (73%; monoallelic 20 of 25, biallelic 40 of 57) exhibiting drug resistance. Patients with epilepsy met the criteria for developmental and epileptic encephalopathy. Among the 16 patients without epilepsy, 4 had dominant pathogenic variants (3 with n.35A>G, ages 29 months–4 years; 1 with n.4G>A, age 6 years). Twelve (ages 2.6–39 years) had biallelic variants, including three with n.40C>T. One terminated fetus (n.174A>C/n.176G>C) had a polymalformation syndrome.

Clinical presentations varied within each group (n.4G>A, n.35A>G and biallelic variants) but, overall, the clinical spectrums were similar, with no clear genotype–phenotype correlations (Table 1 and Extended Data Fig. 7). Myoclonic seizures were more prevalent in biallelic than monoallelic cases (30 of 40, 75% versus 7 of 24, 29%; two-sided Fisher's test *P* = 0.028). Febrile seizures occurred in biallelic (6 of 52, 12%) and monoallelic (11 of 26, 42%) cases but were preponderant in n.4G>A cases (11 of 16, 69%), and absent in n.35A>G carriers (0 of 10, *P* = 0.027; two-sided Fisher's test). Movement disorders seemed more frequent in biallelic (26 of 53, 49%) than in dominant (3 of 23, 13%) cases (not significant after correction). The most severe phenotypes, including all eight reported deaths (ages 3–19 years; median 15 years) due to respiratory, infectious or epilepsy-related complications, were restricted to biallelic patients. Despite intragroup variability, ID severity, seizures and movement disorders were consistent within families, suggesting that variant combinations contribute to phenotypic variability of recessive NDDs. Minor dysmorphic features, such as broad forehead, midface hypoplasia, a large open mouth, a small chin and down-slanting palpebral fissures were common in patients with available photographs (Fig. 4b,c).

Heterozygous parents of patients with biallelic variants were generally unaffected, except in five cases: a mother with n.5C>T and a father with n.32T>C had childhood epilepsy; a mother with n.61C>T had epilepsy; both parents carrying n.17_35dup were affected, the father with ID and the mother with ID and epilepsy.

Seven patients with monoallelic de novo variants other than n.4G>A and n.35A>G had highly variable phenotypes: four patients had mild ID without epilepsy (*n* = 3) or moderate ID with epilepsy (*n* = 1); one died at 6 days from status epilepticus; one fetus was terminated due to semi-lobar holoprosencephaly. The remaining patient had hearing loss and chronic bronchopulmonary disease.

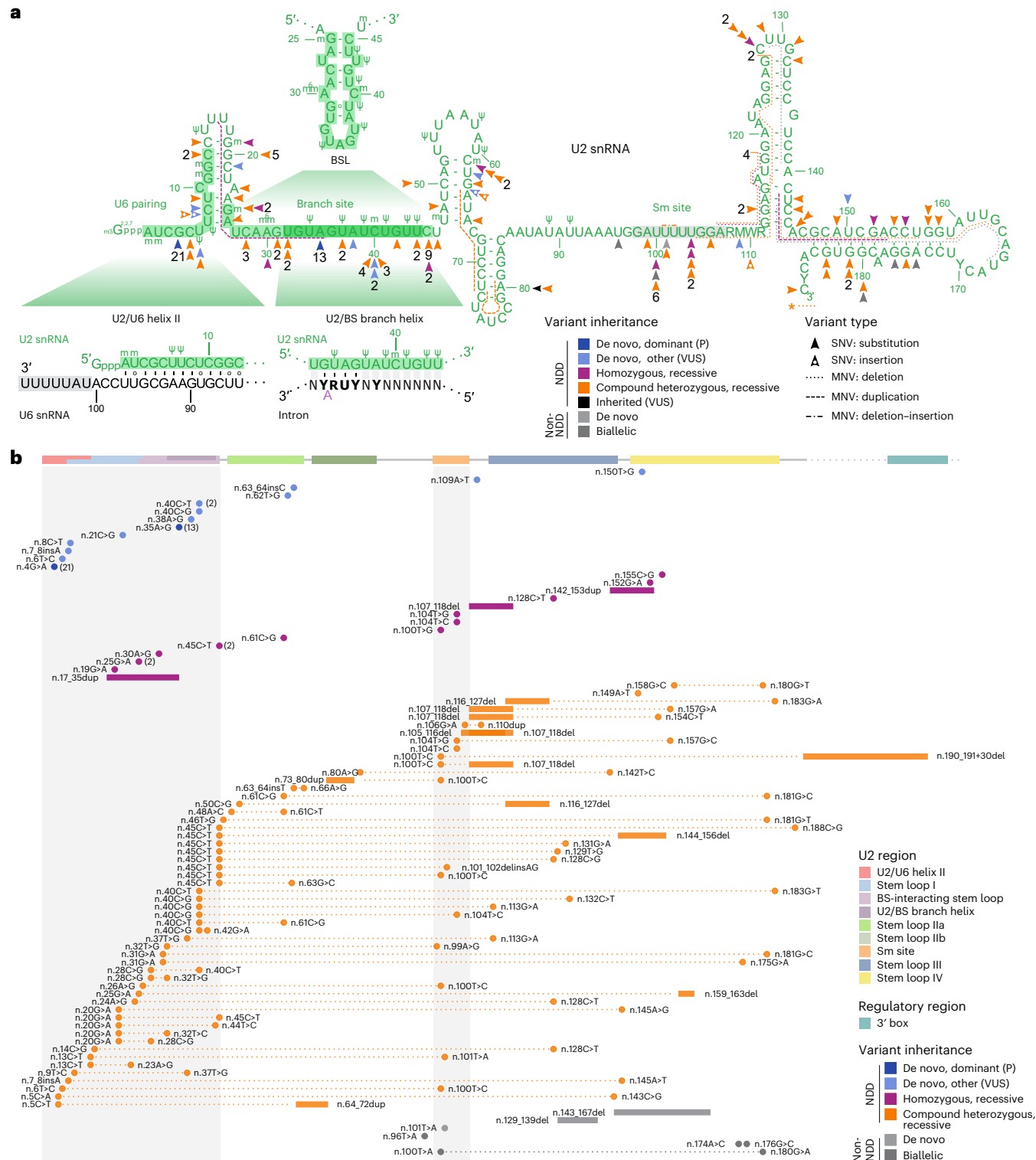

**Fig. 3 | Overview of *RNU2-2* variants identified in this study. a**, Two-dimensional predicted structure of U2-2 snRNA showing structural and functional domains. Arrowheads indicate point variants identified in this study. Variants are colored according to their inheritance. Dark blue: de novo, dominant (n.4G>A or n.35A>G); light blue: de novo other (VUS); orange: compound heterozygous, recessive; purple: homozygous, recessive. The numbers in black represent the count of patients with each variant, for variants identified in more than one family. Other variant types are shown with dotted (deletions), dashed (duplications) or dotted-dashed (indels) lines. Asterisk, deletion encompassing the 3′ box. The nucleotide differences between *RNU2-2* and *RNU2-1* are shown using IUPAC codes. Green numbers refer to the numbering of U2-2 nucleotides

(nt). Ψ, pseudouridine; m, 2′-O-methyl residues; m6, N6-methyladenosine; [2,2,7]m3Gppp, 2,2,7-trimethylguanosine cap. Green-shaded regions: functional domains of U2 involved in spliceosomal activity: U2/U6 helix II (nt 1–13); BSL (nt 25–45) and U2/BS branch helix (nt 32–44). Gray-shaded region: Sm site. **b**, Locations of variants on the *RNU2-2* gene, with different domains of the snRNA highlighted on the schematic, extending to the 3′ box (regulatory region). Note the clustering of variants in the U2/U6 helix, BSL regions and Sm site (gray-shaded), as well as preferential associations of compound heterozygous variants. Variants are ordered by inheritance mode and by the position of the first variant on the snRNA. Variant coloring as in **a**.

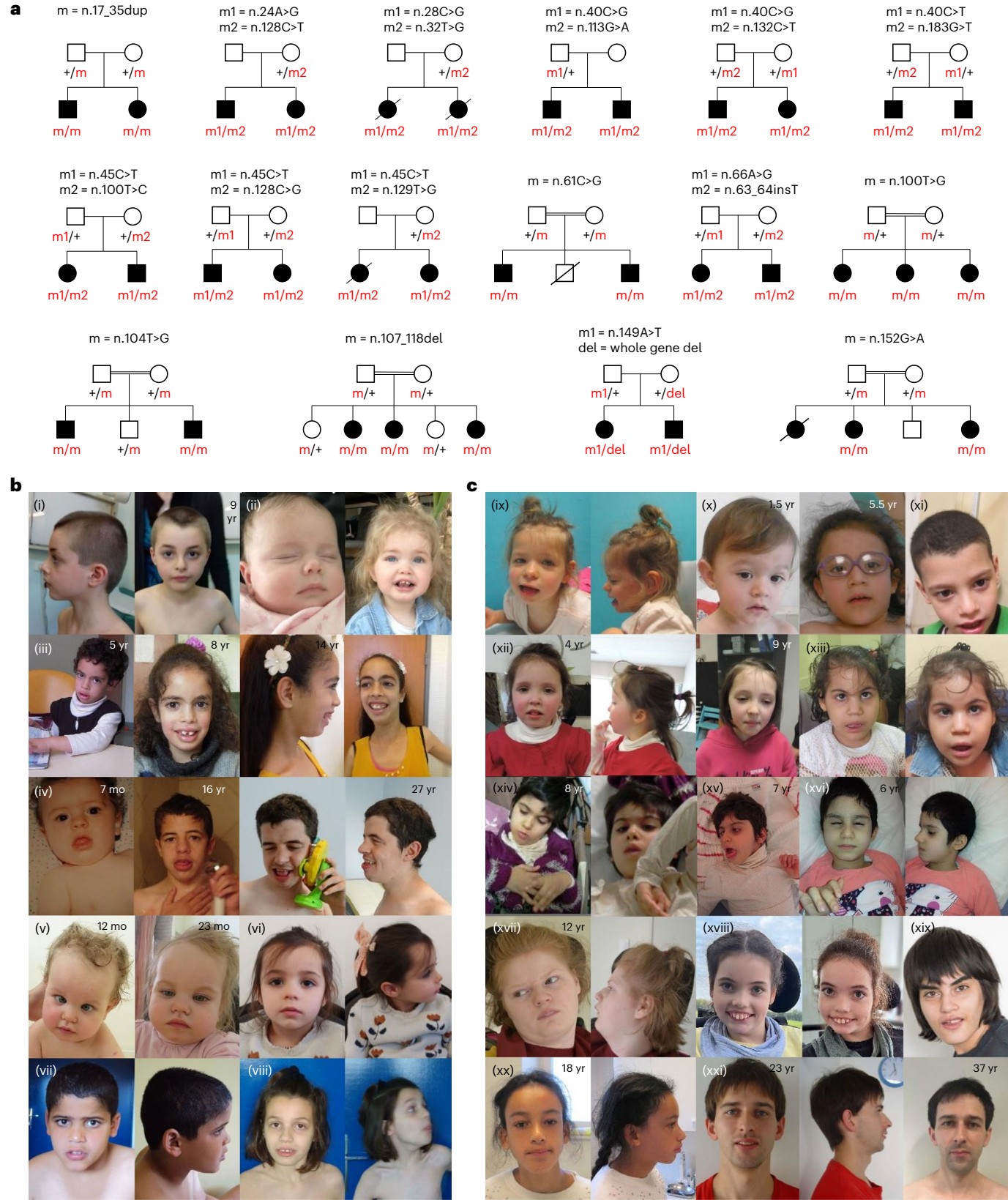

**Fig. 4 | Variant segregation and facial features associated with *RNU2-2* variants. a**, Segregation of biallelic variants in the 16 families with at least two affected siblings. **b**, Facial photographs of people with monoallelic *RNU2-2* variants: (i)–(iv) n.4G>A; (v)–(vii), n.35A>G; (viii) n.35A>G in mosaic. **c**, Facial photographs of individuals with biallelic *RNU2-2* variants. (ix) maternal n.104T>C/paternal gene deletion; (x) maternal n.40C>G/paternal n.42G>A;

(xi) homozygous n.61C>G; (xii) paternal n.48A>C/maternal n.61C>T; (xiii) homozygous n.152G>A; (xiv)–(xvi) homozygous n.107_118del in three affected sisters; (xvii) de novo n.6T>C/maternal n.100T>C; (xviii) maternal n. 20G>A/paternal n.28C>G; (xix) homozygous n.100T>G; (xx) homozygous n.128C>T; (xxi) nonmaternal n.13C>T/maternal n.101T>A. Consent was obtained for publication of facial photographs. mo, months; yr, years.

**Table 1 | Overview of clinical features in patients with monoallelic and biallelic *RNU2-2* variants**

| Group | n.4G>A | n.35A>G | Monoallelic | Biallelic | Total |
|---|---|---|---|---|---|
| **Number of patients** | **20** | **12** | **32** | **73** | **105** |
| Epilepsy | 19 of 20 (95%) | 9 of 12 (75%) | 28 of 32 (88%) | 60 of 72 (83%) | 88 of 104 (85%) |
| Generalized seizures | 9 of 16 (56%) | 6 of 8 (75%) | 15 of 24 (62%) | 39 of 44 (89%) | 54 of 68 (79%) |
| Tonic-clonic seizures | 6 of 15 (40%) | 4 of 8 (50%) | 10 of 23 (43%) | 28 of 37 (76%) | 38 of 60 (63%) |
| Myoclonic seizures | 4 of 15 (27%) | 3 of 9 (33%) | **7 of 24 (29%)** | **30 of 40 (75%)** | 37 of 64 (58%) |
| Tonic seizures | 13 of 16 (81%) | 6 of 8 (75%) | 19 of 24 (79%) | 13 of 32 (41%) | 32 of 56 (57%) |
| Focal seizures | 7 of 15 (47%) | 3 of 9 (33%) | 10 of 24 (42%) | 19 of 31 (61%) | 29 of 55 (53%) |
| Nonmotor seizures (absences) | 9 of 16 (56%) | 1 of 8 (12%) | 10 of 24 (42%) | 22 of 39 (56%) | 32 of 63 (51%) |
| Spasms | 5 of 16 (31%) | 1 of 8 (12%) | 6 of 24 (25%) | 15 of 35 (43%) | 21 of 59 (36%) |
| Atonic seizures | 3 of 15 (20%) | 3 of 8 (38%) | 6 of 23 (26%) | 13 of 33 (39%) | 19 of 56 (34%) |
| Clonic seizures | 7 of 16 (44%) | 1 of 8 (12%) | 8 of 24 (33%) | 9 of 27 (33%) | 17 of 51 (33%) |
| Generalized and focal seizures | 7 of 16 (44%) | 4 of 8 (50%) | 11 of 24 (46%) | 7 of 46 (15%) | 18 of 70 (26%) |
| Hemicorporal seizures | 6 of 15 (40%) | 1 of 8 (12%) | 7 of 23 (30%) | 4 of 27 (15%) | 11 of 50 (22%) |
| Febrile seizures | **11 of 16 (69%)** | **0 of 10 (0%)** | 11 of 26 (42%) | 6 of 52 (12%) | 17 of 78 (22%) |
| Status epilepticus | 15 of 18 (83%) | 5 of 7 (71%) | 20 of 25 (80%) | 40 of 57 (70%) | 60 of 82 (73%) |
| Pharmoresistance | 8 of 17 (47%) | 6 of 8 (75%) | 14 of 25 (56%) | 18 of 51 (35%) | 32 of 76 (42%) |
| Age of seizure onset: <3 years | 15 of 19 (79%) | 7 of 9 (78%) | 22 of 28 (79%) | 50 of 60 (83%) | 72 of 88 (82%) |
| Age of seizure onset: >3 years | 4 of 19 (21%) | 2 of 9 (22%) | 6 of 28 (21%) | 10 of 60 (17%) | 16 of 88 (18%) |
| Daily seizures | 11 of 15 (73%) | 1 of 8 (12%) | 12 of 23 (52%) | 29 of 51 (57%) | 41 of 74 (55%) |
| Severe intellectual disability | 15 of 19 (79%) | 10/10 (100%) | 25 of 29 (86%) | 50 of 68 (74%) | 75 of 97 (77%) |
| Severe developmental delay | 13 of 20 (65%) | 12 of 12 (100%) | 25 of 32 (78%) | 51 of 70 (73%) | 76 of 102 (75%) |
| No language | 15 of 20 (75%) | 8 of 11 (73%) | 23 of 31 (74%) | 41 of 62 (66%) | 64 of 93 (69%) |
| Autism spectrum disorder | 14 of 20 (70%) | 9 of 11 (82%) | 23 of 31 (74%) | 31 of 60 (52%) | 54 of 91 (59%) |
| Hospitalization | 11 of 18 (61%) | 5 of 12 (42%) | 16 of 30 (53%) | 37 of 61 (61%) | 53 of 91 (58%) |
| Stereotypies | 14 of 19 (74%) | 6 of 11 (55%) | 20 of 30 (67%) | 32 of 61 (52%) | 52 of 91 (57%) |
| Other behavioral anomalies | 6 of 17 (35%) | 8 of 11 (73%) | 14 of 28 (50%) | 31 of 58 (53%) | 45 of 86 (52%) |
| Feeding issues, gastro-intestinal reflux | 8 of 18 (44%) | 4 of 12 (33%) | 12 of 30 (40%) | 38 of 67 (57%) | 50 of 97 (52%) |
| Sleep disorders | 11 of 19 (58%) | 1 of 11 (9%) | 12 of 30 (40%) | 37 of 64 (58%) | 49 of 94 (52%) |
| MRI abnormalities | 6 of 18 (33%) | 3 of 10 (30%) | 9 of 28 (32%) | 38 of 63 (60%) | 47 of 91 (52%) |
| Constipation | 9 of 18 (50%) | 3 of 11 (27%) | 12 of 29 (41%) | 31 of 64 (48%) | 43 of 93 (46%) |
| Neonatal hypotonia | 9 of 19 (47%) | 5 of 12 (42%) | 14 of 31 (45%) | 29 of 71 (41%) | 43 of 102 (42%) |
| Dysmorphic features | 9 of 19 (47%) | 6 of 12 (50%) | 15 of 31 (48%) | 24 of 67 (36%) | 39 of 98 (40%) |
| Regression | 5 of 19 (26%) | 4 of 11 (36%) | 9 of 30 (30%) | 28 of 66 (42%) | 37 of 96 (39%) |
| Movement disorder | 3 of 15 (20%) | 0 of 8 (0%) | 3 of 23 (13%) | 26 of 53 (49%) | 29 of 76 (38%) |
| Hypersialorrhea/drooling | 9 of 18 (50%) | 4 of 12 (33%) | 13 of 30 (43%) | 21 of 64 (33%) | 34 of 94 (36%) |
| Short stature | 8 of 20 (40%) | 5 of 11 (45%) | 13 of 31 (42%) | 19 of 63 (30%) | 32 of 94 (34%) |
| Eyes/vision abnormalities | 3 of 18 (17%) | 3 of 12 (25%) | 6 of 30 (20%) | 25 of 64 (39%) | 31 of 94 (33%) |
| Joint hyperlaxity | 5 of 18 (28%) | 3 of 12 (25%) | 8 of 30 (27%) | 22 of 62 (35%) | 30 of 92 (33%) |
| Microcephaly | 4/18 (22%) | 3 of 11 (27%) | 7 of 29 (24%) | 22 of 64 (34%) | 29 of 93 (31%) |
| Ataxia | 7 of 18 (39%) | 5 of 9 (56%) | 12 of 27 (44%) | 11 of 51 (22%) | 23 of 78 (29%) |
| Failure to thrive/growth retardation | 5 of 19 (26%) | 3 of 12 (25%) | 8 of 31 (26%) | 19 of 62 (31%) | 27 of 93 (29%) |
| Bone/skeletal anomalies | 4 of 18 (22%) | 2 of 12 (17%) | 6 of 30 (20%) | 18 of 65 (28%) | 24 of 95 (25%) |
| Hyperventilation | 3 of 18 (17%) | 0 of 11 (0%) | 3 of 29 (10%) | 13 of 64 (20%) | 16 of 93 (17%) |
| Prenatal findings | 1 of 18 (6%) | 2 of 12 (17%) | 3 of 30 (10%) | 9 of 65 (14%) | 12 of 95 (13%) |

MRI, magnetic resonance imaging. Clinical features are listed by frequency across the entire patient cohort, with significant differences between groups shown in bold. Detailed clinical features and statistical comparisons for all three groups, including patients with de novo variants lacking a second pathogenic allele, are shown in Supplementary Table 10 (index cases only) and Supplementary Table 11 (index cases+siblings).

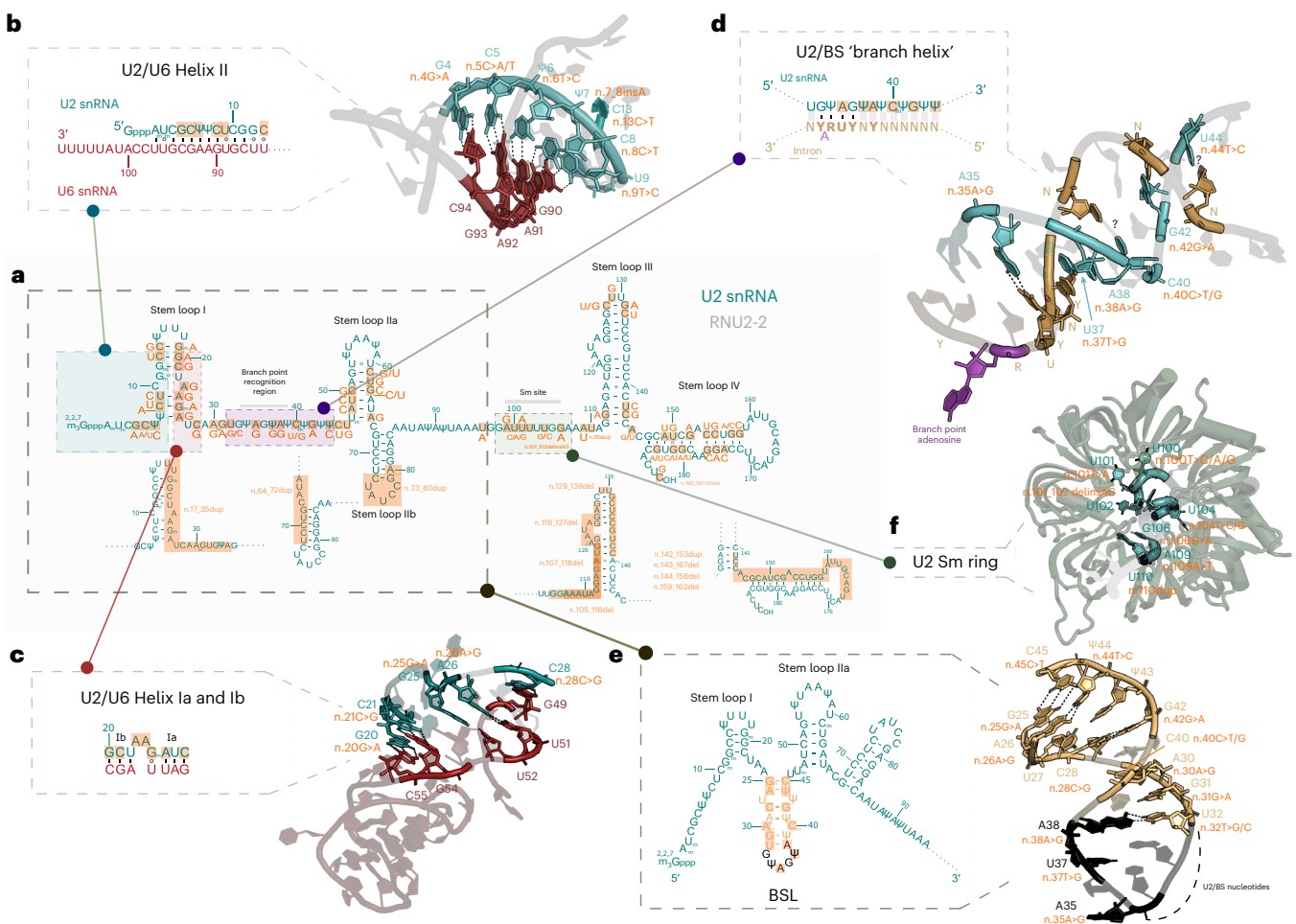

**Fig. 5 | Possible functional impact of *RNU2-2* variants.** Overview of *RNU2-2* variants identified in this study. **a**, Two-dimensional structure of the U2 snRNA (green). Orange boxes indicate variants from this study, with point mutations and single nucleotide insertions represented on the graph along with their respective changes. 2,2,7m3Gppp, 2,2,7-trimethylguanosine cap; Ψ, pseudouridine; m, 2′-O-methyl residues. **b**, Zoom-in box representing the two-dimensional predicted structure of the U2 (teal) and U6 (red) snRNAs U2/U6 helix II during formation of precatalytic spliceosome. Interactions stabilizing this structure as well as mutations potentially affecting its stability are represented on coordindates (PDB 5XJC). **c**, Zoom-in box representing the two-dimensional predicted structure of U2/U6 helices Ia and Ib. Nucleotides involved in the structure and associated with pathological variants are represented (PDB 5XJC).

**d**, Close-up of U2/BS branch helix where the two-dimensional predicted structure of the interaction between branch-point recognition region of U2 snRNA and the intronic BS (light brown) is represented. The branch-point adenosine is highlighted in purple. 'YNYURAY,' consensus BS in metazoans: Y, pyrimidine; N, any nucleotide; R, purine. The mutations in this region are depicted on the structure (PDB 5XJC). **e**, Zoom-in box representing the two-dimensional predicted structure of U2 snRNA within the 17S snRNP with the BSL highlighted. In black are nucleotides involved directly on the BS recognition. The structure of the BSL within the 17S U2 snRNP is represented (PDB 7Q3L). **f**, Close-up of the U2 snRNA Sm ring (teal), with nucleotides with mutations close to, or within, it marked in orange (PDB 5XJC).

## Predicted impact of *RNU2-2* variants

To assess the impact of *RNU2-2* variants, we examined their distribution across U2 structural and functional domains and mapped them onto published U2 snRNP and spliceosome structures[30,31]. U2-2 can be divided into three functional domains, each remodeled to different extents throughout the splicing cycle and snRNP biogenesis. The 5′ domain forms four partially mutually exclusive structures: (1) the intramolecular branch-point-interacting stem–loop (BSL; n.25–45)[32], (2) stem–loop I (SLI; n.7–26)[33], (3) the intermolecular U2/U6 helix II (n.1–13)[1] and (4) the U2/U6 helix Ia and Ib (n.20–28)[34]. The branch interacting region oscillates between BSL conformation and stable intron binding through formation of the U2/BS branch helix (n.32–44)[1,35]. The 3′ end domain encompasses five structural elements: stem loops IIa (SLIIa; n.47–66)[36] and IIb (SLIIb; n.68–84)[32], the Sm binding site (n.98–107)[37] and 3′ stem loops III and IV (SLIII; n.112–144/SLIV; n.147–184)[38,39].

*RNU2-2* variants at the 5′ end probably destabilize U2/U6 helix II by disrupting key Watson–Crick base pairs[33]: n.4G>A breaks the G4–C94

pair, n.5C>A/T disrupts C5–G93, n.6T>C alters U6–A92 and n.8C>T the C8–G90 (Fig. 5). These changes may impair tri-snRNP recruitment to the prespliceosome and reduce splicing efficiency[40–42]. n.7_8insA, n.8C>T, n.9T>C, n.13C>T, n.14C>G, n.19G>A, n.20G>A, n.21C>G, n.23A>G, n.24A>G, n.25G>A and n.26A>G would affect SL1 stability and directly or indirectly alter U2/U6 helix II formation. Furthermore, n.20G>A, n.21C>G, n.26A>G and n.28C>G may destabilize helices Ia and Ib by replacing Watson–Crick base pairs with noncanonical pairs, perturbing active site formation.

Some variants within the BSL probably affect its stability. For instance, n.40C>T or n.30A>G may create a Watson–Crick base pair between positions 30 and 40 that hyperstabilizes the BSL, which was linked previously to reduced splicing fidelity in yeast[32]. Likewise, n.40C>G may stabilize the BSL by creating a G–A base pair replacing the C–A mismatch[43]. Conversely, n.25G>A, n.26A>G, n.28C>G, n.32T>C/G, n.38A>G, n.42G>A, n.44T>C and n.45C>T may disrupt BSL integrity[44]. Notably, n.35A>G alters the invariant A35 of U2 that pairs with the conserved U upstream of the branch site (BS), which may increase pairing

flexibility, promoting cryptic BS usage[45] (Fig. 5). Besides, n.28C>G, which may affect both BSL formation and the active site, is a recurrent somatic variant detected in cancers[24].

In contrast, variants in SLIIa, the Sm site and the 3′ stem loops SLIII/SLIV probably impair U2 snRNP biogenesis and nuclear import, destabilizing U2 and preventing proper spliceosome assembly. Likewise, n.190_191+30del would perturb proper 3′ end processing. These changes may behave effectively as null alleles such as gene deletions, as the affected U2 molecules may fail to reach functional spliceosomes.

### Impact of *RNU2-2* variants on lymphocyte transcriptome

We showed previously that pathogenic *RNU4-2* variants lead to splicing defects affecting primarily alternative 5′SS, detectable in cultured lymphocytes[18]. Here we investigated splicing alterations in 19 people with either dominant (n.4G>A, *n* = 5; n.35A>G, *n* = 5) or recessive (*n* = 9) *RNU2-2* variants and compared them with data of 49 patients with other disorders (controls). Using the same approach as for *RNU4-2*, we found no consistent splicing signature among *RNU2-2* variant carriers. To increase the specificity of our analysis, we applied a linear regression on percent-spliced-in (PSI) values (ΔPSI > 0.05), integrating age, sex and cell composition as covariates. Significant alternative splicing events (*P* < 0.01) were detected for 5′SS, alternative 3′ splice sites (3′SS), skipped exons (SEs) and retained introns (RIs). SEs represented the main effect in both dominant (n.4G>A and n.35A>G) and recessive *RNU2-2* (Fig. 6a) cases but showed minimal overlap between all three conditions (Fig. 6b). Clustering of PSI residuals for SE events distinguished *RNU2-2* variant carriers successfully from controls (Fig. 6c and Supplementary Tables 12–14). Analysis of ΔPSI distributions revealed a global decrease in exon inclusion (that is, increased exon skipping) in dominant cases, whereas recessive cases exhibited both increased and decreased exon skipping to a similar extent (Fig. 6d). All SE events were also present among controls (Fig. 6e), indicating that *RNU2-2* variants induce subtle quantitative perturbations in alternative splicing rather than generating new splice junctions in lymphocytes.

### Impact of *RNU2-2* variants on blood methylome

We next analyzed DNA methylation profiles in 24 patients with *RNU2-2* variants (8 with n.4G>A, 6 with n.35A>G and 10 with biallelic variants) and compared them with 68 controls to identify potential episignatures, using a similar methodology to *RNU4-2*[18]. Cases and controls were matched for age at sampling, and analyses also corrected for age, sex and cell composition. Consistent with transcriptomic findings, patient groups seemed heterogeneous, limiting statistical power for variant-specific analyses. We then implemented a combined model simultaneously evaluating all three variant types while allowing for both shared and variant-specific effects. Overall, methylation signatures associated with *RNU2-2* variants were subtle, heterogeneous and weaker than reported previously for *RNU4-2* (Fig. 7a). Principal component (PC) analysis (PCA) across all 201 differentially methylated positions showed that n.4G>A carriers exhibited the strongest signal, although only ~25% of variance was captured by PC1 (Extended Data Fig. 8a and Supplementary Table 15). Separation of n.35A>G and biallelic carriers from controls became apparent only at higher PCs, reflecting the minor effect size. Excluding n.4G>A carriers improved visualization, with n.35A>G and biallelic variants separating partially from controls along PC2 (Extended Data Fig. 8b,c). Methylation profiles showed that n.4G>A carriers were largely hypomethylated, whereas n.35A>G carriers displayed modest hypermethylation (~2.5% Δβ) and biallelic carriers had heterogeneous patterns without consistent hypo- or hypermethylation (Fig. 7b). Cross-validation with a single four-class support vector machine (SVM) classifier confirmed the strongest signature for n.4G>A (sensitivity 100%), with lower performance for n.35A>G (83%) and biallelic (80%) variants and an overall specificity of 87% (Extended Data Fig. 8d). Overall, methylation differences

between variant carriers and controls were less than 5% Δβ, near the technical detection limit, indicating subtle and heterogeneous effects across variants.

## Discussion

Although historically overlooked due to their high conservation and redundancy, variants in snRNAs are now recognized as principal contributors to genetic disorders. In this study, we analyzed all annotated snRNA genes, hypothesizing that some 'pseudogenes' may be functional disease genes. Our main finding was the identification of biallelic variants in *RNU2-2* as the cause of a frequent recessive NDD. Clinically, this recessive NDD closely resembles the recently described dominant *RNU2-2*–associated disorder caused by n.4G>A and n.35A>G[19,23], but seems to be at least twice as common. The clinical phenotype associated with dominant or biallelic *RNU2-2* variants is characterized by neurodevelopmental delay, severe ID and epilepsy in 85% of cases, usually starting before 3 years of age. Excluding monoallelic de novo variants other than n.4G>A and n.35A>G, we estimate that dominant and recessive *RNU2-2* variants altogether account for at least ~0.35% of NDDs (Supplementary Table 7), similar to the prevalence of ReNU syndrome[16,18]. Interpreting *RNU2-2* variants is complicated by the relatively high and overlapping allele frequencies of pathogenic and benign variants. It is thus possible that a few biallelic variants reported in our study are nonpathogenic, whereas others not considered could contribute to disease or modify penetrance (like for example, n.99A>G). Compared to *RNU4-2*, which is associated mainly with dominant ReNU syndrome, *RNU2-2* shows lower constraint and greater tolerance to variation, consistent with its association with predominant recessive disorders. snRNA genes are hypermutable and saturated with variants[16,25], probably explaining the discrepancy between their small size and the high frequency of the associated phenotypes.

The identification of *RNU2-2* biallelic variants as a common cause of recessive disorders was replicated independently by two groups[46–48]. Both other studies relied mainly on Genomics England data and report largely the same pathogenic variants, whereas our work is based on the PFMG and other independent cohorts, with no (or negligible) patient overlap. Despite this, the results are remarkably consistent, differing mainly in their interpretation. Unlike previous studies from the same authors, which reported de novo variants separately[19,23], our data show that de novo variants in snRNA genes can underlie both recessive and dominant inheritance, reflecting their high mutability[25]. Therefore, a second pathogenic variant in trans should always be sought for when evaluating de novo variants in snRNA genes.

Jackson et al.[46] suggest distinct mechanisms and clinical features for dominant versus recessive *RNU2-2* variants, whereas our detailed clinical analysis of >100 patients, combined with structural predictions of variant impacts, rather indicates a continuum between dominant and recessive disorders. *RNU2-2* exhibits a broad and complex spectrum of variant effects that blurs the lines between dominant and recessive inheritance. De novo variants near n.4G>A and n.35A>G are particularly hard to interpret in the absence of a second pathogenic allele. These variants are extremely rare or absent from population databases (that is, under stronger negative selection than other biallelic variants) and recur de novo across unsolved patients with NDD. Together, these findings suggest that certain heterozygous variants could contribute to dominant disease with variable penetrance, yet produce a classic recessive phenotype when paired with a second pathogenic allele in *trans*.

These observations must be interpreted in light of predicted variant effects on U2-2 snRNA and the presence of additional U2 copies, including canonical U2-1 (*RNU2-1*) paralogs. U2 contributes largely to defining splicing outcomes by selecting the intronic BS (consensus 'YNYURAY' in metazoans) and thereby the acceptor splice sites[49]. Before its integration within the spliceosome, U2 snRNAs undergo a maturation process that includes Sm-core assembly, nuclear reimport and protein recruitment to form the U2 snRNP[38,39,50–53]. The 3′ domain

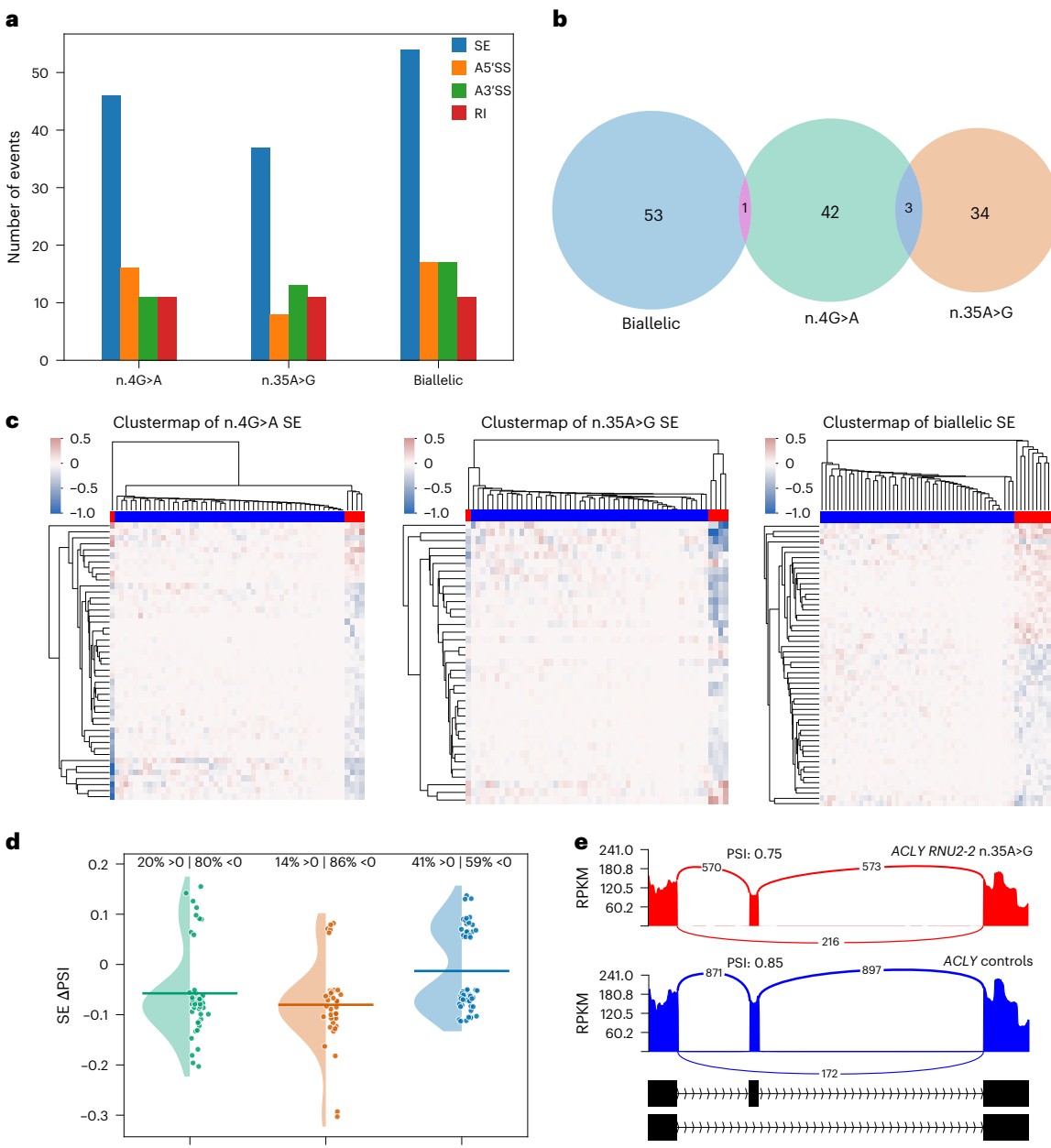

**Fig. 6 | Variant-specific alternative splicing perturbations in *RNU2-2* variant carriers. a**, Number of significant alternative splicing events (ΔPSI > 0.05) detected using rMATS-turbo and linear regression (*P* < 0.01), comparing *RNU2-2* patients (dominant: n.4G>A, *n* = 5; n.35A>G, *n* = 5; biallelic: *n* = 9) to 49 controls. Splicing categories are color-coded: SE (blue), alternative 5′SS (A5′SS, orange), alternative 3′SS (A3′SS, green) and RI (red). **b**, Venn diagram showing minimal overlap of exon skipping events shared among dominant (n.4G>A, n.35A>G) and recessive (biallelic) variant groups. **c**, Clustermaps of PSI value residuals after linear regression for SE events of patients carrying n.4G>A (left, *n* = 5),

n.35A>G (middle, *n* = 5) and biallelic (right, *n* = 9) variants. Blue: controls; red: *RNU2-2* cases. **d**, Distribution of ΔPSI across the three conditions. Raincloud plots show the density of ΔPSI values (half-violins), individual splicing events (points) and mean ΔPSI (horizontal bars). Percentages above each group indicate the proportion of events with increased (ΔPSI > 0) or decreased (ΔPSI < 0) exon inclusion. **e**, Sashimi plots illustrating an SE isoform shift in *ACLY* observed in *RNU2-2* n.35A>G variant carriers compared to controls. RPKM, reads per kilobase of transcript per million mapped reads.

mediates structural interactions necessary for U2 snRNP biogenesis and spliceosome assembly[38,53,54], such as SLIIa binding SF3B1[50,55]. In contrast, the 5′ domain contributes to spliceosome assembly, active site formation and branch-point recognition through highly conserved RNA–RNA interactions. During early spliceosome assembly, U2 initially probes the BS and forms a stable U2/BS branch helix[32]. This process requires the PRP5 ATPase, which remodels the BSL and enforces fidelity of BS recognition[40]. Formation of the branch helix may trigger SLI formation[55], which is disrupted subsequently to allow tri-snRNP recruitment through creation of the U2/U6 helix II, to form

the precatalytic spliceosome. Upon spliceosome activation, U2 forms two additional contiguous helices with U6 (Ia, Ib) that, along with helix II, flank and stabilize the RNA-based active site[56,57]. Variants in the Sm site or 3′ stem loops probably prevent U2 snRNP maturation and spliceosome incorporation. Mutant U2-2 RNAs that fail to mature properly behave like null alleles, similar to larger indels or whole-gene deletions. These null alleles are tolerated when heterozygous but pathogenic in homozygous or compound heterozygous states, highlighting the essential role of *RNU2-2* in human brain development. In contrast, variants in the U2-2 5′ region, spanning the U2/U6 helix II and

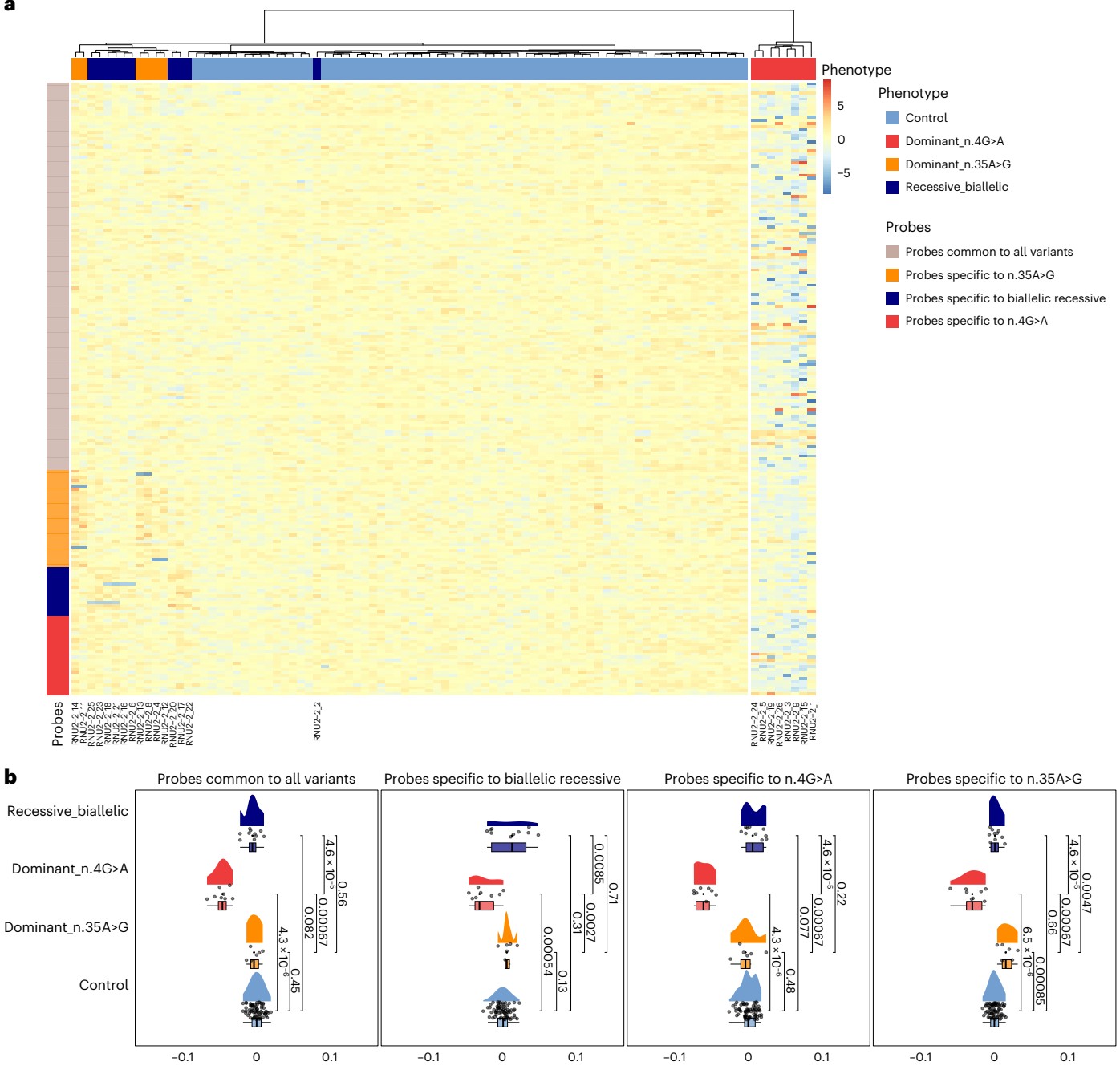

**Fig. 7 | Methylation profiles across differentially methylated probes in _RNU2-2_ variant carriers and normal controls.** In all panels, controls (_n_ = 68) are shown in light blue, dominant n.4G>A carriers (_n_ = 8) in red, dominant n.35A>G carriers (_n_ = 6) in orange and recessive biallelic cases (_n_ = 10) in dark blue. **a**, Heatmap of adjusted methylation levels at differentially methylated CpGs. Selected probes are represented in rows whereas patients and normal controls are represented in columns after hierarchical clustering of their methylation profiles. Probes are grouped according to their association pattern: common to all variants, specific to n.35A>G, specific to biallelic recessive variants and specific to n.4G>A.

Samples are annotated by type (control, dominant n.4G>A, dominant n.35A>G, recessive biallelic) but sample clustering was unsupervised and blind to this classification. **b**, Raincloud plots showing the average methylation level per sample within each probe association pattern. Normal controls and variant carriers display significantly different distributions, with _P_ values from two-sided Wilcoxon rank-sum tests indicated on the plots. The number of individuals per group is the same as in **a**. Boxplot elements are as follows: centerline, median; box limits, upper and lower quartiles; whiskers, 1.5× interquartile range.

BSL, are predicted to be incorporated into spliceosomes, although some may also alter U2-2 snRNA levels[58]. Recurrent variants n.4G>A and n.35A>G seem sufficient to reach the pathogenic threshold in the heterozygous state, whereas nearby variants are typically insufficient alone but can cause disease when paired with another variant in _trans_. This corresponds to a gradient-of-impact framework, where _RNU2-2_ variants (1) can act dominantly on their own (n.4G>A; n.35A>G); (2)

require a second nonfunctional allele to cause disease (one non-null allele, one null allele); (3) are both incorporated in spliceosomes, having additive effects (two non-null alleles) or (4) fail to enter the spliceosome entirely (two null alleles). Jackson et al. suggested that the U2-2:U2-1 ratio could serve as a biomarker for _RNU2-2_ recessive disorders[46]. However, it is possible that this biomarker is valid only when at least one variant impairs U2-2 maturation.

Contrary to *RNU4-2*, *RNU2-2* variants exert a subtler effect on splicing in blood, causing primarily exon skipping for dominant variants and broader splicing alterations for recessive variants. These findings align with the lower expression of *RNU2-2* than *RNU2-1* in blood, which contrasts with its higher relative expression in the brain[19,23], and is consistent with the primarily neurodevelopmental features of *RNU2-2*–associated disorders. Similarly, DNA methylation analyses in *RNU2-2* variant carriers revealed only modest, variant-specific changes. These results should be interpreted cautiously and validated in larger cohorts, as some differences may reflect stochastic variability rather than consistent, biologically meaningful episignatures, and may also mirror the variants' diverse functional effects.

Despite the size of the PFMG cohort (>34,000 participants), we did not identify additional significant snRNA-disease associations. This probably reflects limited statistical power to detect variants in snRNA genes causing rarer disorders, as known disease associations with de novo variants in *RNU5B-1*[18,19] or recessive variants in *RNU4-2*[21,22], *RNU-6ATAC*[59] and *RNU12*[14,15] did not reach significance. Increasing statistical power will require pooling international cohorts for robust genotype–phenotype association studies. Another important limitation of our study is the reliance on short-read genome sequencing, which cannot accurately resolve highly similar snRNA loci, including most U1 and U2 canonical genes, due to the presence of (nearly) identical copies contained in regions spanning >5–10 kb in sequence homology. The analysis of these regions thus requires long-read sequencing technologies.

In conclusion, our results support a continuum of dominant and recessive NDDs caused by *RNU2-2* variants, which altogether are nearly as prevalent as ReNU syndrome. They also underscore the challenges of interpreting variants, especially those occurring de novo, in these highly mutable loci, and suggest that additional snRNA-associated disorders remain to be discovered.

## Online content

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

Elsa Leitão [1,132], Amandine Santini [2,132], Benjamin Cogne [3,4,5,132], Miriam Essid [6,7,8], Maria Athanasiadou [9,10], Christy W. LaFlamme [11,12], Pierre Marijon [5], Virginie Bernard[8], Kevin Jousselin[5], Nicolas Chatron [6,7,8], Giulia Barcia[5,13,14], Boris Keren[5,15], Cyril Mignot[15,16], Perrine Charles[15], Thomas Besnard [3,4], Robin Paluch [1], Jean-Madeleine de Sainte Agathe [5,15], Edith P. Almanza Fuerte [11], Soham Sengupta [11], Mathieu Milh[17,18], Francis Ramond [19], Talia Allan [20], Isabelle An[21], Camila Araujo[22], Stéphanie Arpin[23,24], Christina Austin-Tse[25], Stéphane Auvin[26,27], Sarah Baer[28], Nadia Bahi-Buisson[14], Mads Bak [29], Magalie Barth[30], Stéphanie Baulac [31], Nathalie Bednarek-Weirauch[32,33,34], Matthias Begemann [35], Mark F. Bennett[20,36,37], Uriel Bensabath[15], Stéphane Bézieau [3,4], Rakia Bhouri[38], Margaux Biehler[39], Trine Bjørg Hammer[29,40], Julie Bogoin[15], Emilie Bonanno [5,15], Simon Boussion[41], Céline Bris[5,30,42], Adelaide Brosseau-Beauvir[43], Ange-Line Bruel [44], Audrey Briand-Suleau[5],

Julien Buratti [15], Tristan Celse[8,45], Pascal Chambon[2,5], Nicole Chemaly[14,46], Bertrand Chesneau [47], Estelle Colin [30,42], Maxime Colmard[48], Cindy Colson[41], Solène Conrad[3], Thomas Courtin[13,14], Isabelle Creveaux[8,49], Anne-Charlotte Cullier[50], Louis T. Dang [51], Anne de Saint Martin[28], Caroline de Vanssay de Blavous Legendre[52], Bénédicte Demeer[53,54], Anne-Sophie Denommé-Pichon [44], Philine Diekhoff[55], Stephanie DiTroia[25], Martine Doco-Fenzy[3,33,56], Christèle Dubourg [5,57,58], Charlotte Dubucs [59], Stéphanie Ducreux [5,13], Louis Dufour[26], Romain Duquet[15], Benjamin Durand [60], Salima El Chehadeh [60], Miriam Elbracht [35], Laurence Faivre [61], Marie Faoucher[5,8,57,58], Anne Faudet[15], Sylvie Forlani[31], Mélanie Fradin [62], Pauline Gaignard[5,63], Benjamin Ganne [8,64], Aurore Garde [65], Justine Géraud[66], Deepak Gill [67,68,69], Alice Goldenberg[2], David Grabli[21,31], Coraline Grisel[70], Sophie Gueden[71], Paul Gueguen[5,23,24], Anne-Marie Guerrot[2], Agnès Guichet[5,30,42], Tobias B. Haack [72], Nina Härting[1], Martin Georg Häusler[73], Solveig Heide[5,15], Theresia Herget[55], Bénédicte Héron[74], Delphine Héron[15,16], Johanna Herwig [55], Mathilde Heulin[75], Tess Holling [55], Clara Houdayer [30,42], Bertrand Isidor[3], Aurélia Jacquette [76], Louis Januel[6,8], Nolwenn Jean-Marçais[62], Frank J. Kaiser[1], Sabine Kaya[1], Chontelle King[77], Marina Konyukh[5,78], Florian Kraft [35], Jeremias Krause [35], Rémi Kirstetter [13,14], Alma Kuechler[1], Ingo Kurth [35], Kerstin Kutsche[55], Audrey Labalme[6], Jean-Serene Laloy[5,13], Vincent Laugel[28], Floriane Le Bricquir[79], Anne-Sophie Lèbre [33,80,81], Marine Lebrun[8,19], Eric Leguern[15,31], Jonathan Levy [5,82], Nico Lieffering[77], Stanislas Lyonnet[13,14], Kevin Lüthy [1], Sian M. W. Macdonald[20], Lamisse Mansour-Hendili[5], Julien Maraval [61], Iris Marquardt[83], Carolin Mattausch [1], Sandra Mercier[3,4], Olfa Messaoud [25,84,85,86], Godelieve Morel [87], Jérémie Mortreux [8], Arnold Munnich[14], Rima Nabbout[14,46], Sophie Nambot [65], Vincent Navarro[21,31], Ashana Neale[25], Laetitia Nguyen[15], Mathilde Nizon[3], Frédérique Nowak[88], Melanie C. O'Leary [25], Sylvie Odent [58,62], Naomi Meave Ojeda[89,90], Valérie Olin[15], Simone Olivieri[72], Katrin Õunap [91,92], Lynn S. Pais[25], Eleni Panagiotakaki [93,94], Olivier Patat [47], Laurence Perrin-Sabourin[82], Florence Petit[41], Christophe Philippe [8,95], Amélie Piton [8,39,96], Marc Planes[97], Céline Poirsier[56], Antoine Pouzet[82], Clément Prouteau[30,42], Sylvia Quéméner-Redon[5,43,97,98], Mathilde Renaud[99], Anne-Claire Richard[2], Marlène Rio [13,14], Clotilde Rivier[100], Florence Robin-Renaldo[74], Paul Rollier [58,62], Massimiliano Rossi [6,94], Agathe Roubertie[48,101], Valentin Ruault[102], Maïlys Rupin-Mas[71], Pascale Saugier-Veber[2,5], Aline Saunier[95], Russell Saneto[103], Elisabeth Sarrazin[104], Catherine Sarret[105], Elise Schaefer[60], Caroline Schluth-Bolard[8,39,106], Amy Schneider [20], Isabell Schumann[107,108], Vladimir B. Seplyarskiy [109,110], Stephanie Spranger[111], Thomas Smol[5,41], Marc Sturm [72], Shamil R. Sunyaev [109,110], Brian Sperelakis-Beedham[5,13,14], Sarah L. Stenton [25,85], Friedrich Stock[1], Mylène Tharreau[112], Deniz Torun[113], Joseph Toulouse[93], Harshini Thiyagarajah[20], Stéphanie Valence [74], Sophie Valleix[5,114], Julien Van-Gils [115,116], Laurent Villard[117,118], Dorothée Ville[119], Nathalie Villeneuve[17], Antonio Vitobello[8,44], Aurélie Waernessyckle[15], Jan Wagner [120], Yvonne Weber [121], Dagmar Wieczorek [122], Tom Witkowski[20], Manya Yadavilli[89,90], Tony Yammine[80], Khaoula Zaafrane-Khachnaoui[123], Maha S. Zaki [124], Alban Ziegler[8,47], Nuria C. Bramswig [107], Alban Lermine[5], Gael Nicolas [2,5], Joseph G. Gleeson [89,90], Lynette G. Sadleir[77], Michael S. Hildebrand [20,125], Ingrid E. Scheffer[20,126,127], Nicola Whiffin[25,128,129], Anne O'Donnell-Luria [25,84,85,86], Heather C. Mefford [11], Pierre Blanc[5], Julien Thevenon[8,45,130], Camille Charbonnier [131], Clément Charenton[9,10], Christel Depienne [1,5,133] ✉, Gaetan Lesca [6,7,8,133] ✉ & Caroline Nava [5,15,31,133] ✉

[1]Institute of Human Genetics, University Hospital Essen, University Duisburg-Essen, Essen, Germany. [2]Université Rouen Normandie, Normandie Université, Inserm U1245 and CHU Rouen, Department of Genetics and Reference Center for Developmental Abnormalities, Rouen, France. [3]Nantes Université, CHU de Nantes, Service de Génétique Médicale, Nantes, France. [4]Nantes Université, CHU de Nantes, CNRS, INSERM, l'Institut du Thorax, Nantes, France. [5]Laboratoire SeqOIA, Paris, France. [6]Genetics Department, Hospices Civils de Lyon, Lyon, France. [7]Pathophysiology and Genetics of Neuron and Muscle (PNMG), UCBL, CNRS UMR5261 - INSERM, Lyon, France. [8]GCS AURAGEN, Lyon, France. [9]CNRS, Inserm, Université de Strasbourg, IGBMC UMR 7104-UMR-S 1258, Illkirch, France. [10]Department of Integrated Structural Biology, IGBMC, Illkirch, France. [11]Center for Pediatric Neurological Disease Research, St. Jude Children's Research Hospital, Memphis, TN, USA. [12]Graduate School of Biomedical Sciences, St. Jude Children's Research Hospital, Memphis, Memphis, TN, USA. [13]Assistance Publique - Hôpitaux de Paris (APHP), Service de Médecine Génomique des Maladies Rares, Hôpital Necker-Enfants malades, Paris, France. [14]Université Paris Cité, INSERM, IHU Imagine – Institut des maladies génétiques, Paris, France. [15]Département de Génétique Médicale, Assistance Publique - Hôpitaux de Paris (APHP) Sorbonne Université, Hôpital Pitié-Salpêtrière, Paris, France. [16]Centre de Référence Déficiences Intellectuelles de Causes Rares, Paris, France. [17]Service de Neurologie Pediatrique, AP-HM, Marseille, France. [18]Aix Marseille Université, Inserm, INMED, Marseille, France. [19]Département de Génétique, Centre Hospitalier Universitaire de Saint-Etienne, Saint-Etienne, France. [20]Department of Medicine, Epilepsy Research Centre, University of Melbourne, Austin Health, Heidelberg, Victoria, Australia. [21]Département de Neurologie, Assistance Publique - Hôpitaux de Paris (APHP) Sorbonne Université, Center of Reference for Rare Epilepsies, ERN EPICARE, Hôpital Pitié-Salpêtrière, Paris, France. [22]Department of Surgery and Anatomy, Ribeirão Preto Medical School, University of São Paulo, Ribeirao Preto, Brazil. [23]Service de Génétique, CHU de Tours, Tours, France. [24]Université de Tours, INSERM, Imaging Brain and Neuropsychiatry iBraiN U1253, Tours, France. [25]Broad Center for Mendelian Genomics, Program in Medical and Population Genetics, Broad Institute of MIT and Harvard, Cambridge, MA, USA. [26]Département de Neuropédiatrie, Assistance Publique - Hôpitaux de Paris (APHP), Hôpital Robert-Debré, Paris, France. [27]Université Paris Cité, INSERM NeuroDiderot, Paris, France. [28]Service de Neuropédiatrie, Hôpitaux Universitaires de Strasbourg, Strasbourg, France. [29]Department of Clinical Genetics, Copenhagen University Hospital, Rigshospitalet, Denmark. [30]Department of Medical Genetics, Angers University Hospital, Angers, France. [31]Institut du Cerveau - Paris Brain Institute - ICM, Sorbonne Université, Inserm, CNRS, APHP, Hôpital de la Pitié Salpêtrière, Paris, France. [32]Service de Pédiatrie, CHU Reims, Reims, France. [33]Université Reims Champagne Ardenne (URCA), UFR médecine, Reims, France. [34]CReSTIC/EA 3804, URCA, Reims, France. [35]Center for Human Genetics and

Genomic Medicine, Medical Faculty, RWTH Aachen University Hospital, Aachen, Germany. [36]Genetics and Gene Regulation Division, Walter and Eliza Hall Institute of Medical Research, Parkville, Victoria, Australia. [37]Department of Medical Biology, The University of Melbourne, Parkville, Victoria, Australia. [38]Service de Génétique Médicale, Centre Hospitalier Intercommunal de Créteil (CHIC), Créteil, France. [39]Laboratoire de Diagnostic Génétique, Nouvel Hôpital Civil, Hôpitaux Universitaires de Strasbourg, Strasbourg, France. [40]Danish Epilepsy center, Dianalund, Denmark. [41]CHU Lille, Université Lille, ULR7364 – RADEME, Lille, France. [42]MitoLab, Unité MITOVASC, UMR CNRS 6015, INSERM U1083, SFR ICAT, University Hospital of Angers, Angers, France. [43]Center for Intellectual Disability Reference, Brest University Hospital, Brest, France. [44]Laboratoire de Génomique Médicale, Centre Neomics, FHU-TRANSLAD, Centre de Recherche Translationnelle en Médecine Moléculaire – Inserm UMR1231 équipe GAD, Université Bourgogne Europe, CHU Dijon Bourgogne, Dijon, France. [45]Service de Génétique, Génomique et Procréation, CHU Grenoble Alpes, Grenoble, France. [46]Department of Pediatric Neurology, Reference Center for Rare Epilepsies, Necker Enfants Malades Hospital, Paris, France. [47]Service de Génétique Médicale, CHU Purpan, Toulouse, France. [48]Service de Neuropédiatrie, CHU Montpellier, Montpellier, France. [49]Laboratoire de Génétique Moléculaire, CHU de Clermont-Ferrand, Clermont-Ferrand, France. [50]Service de Pediatrie, CHR Metz-Thionville, Hôpital Mercy, Metz, France. [51]Department of Pediatrics, Michigan Medicine, University of Michigan, Ann Arbor, MI, USA. [52]Service de Pédiatrie, Consultation de Neurologie Pédiatrique GHH Jacques Monod, Le Havre, France. [53]Service de Génétique Clinique et Oncogénétique, CLAD Nord-ouest, CHU Amiens-Picardie, Amiens, France. [54]CHIMERE - INSERM UA-21, Université Picardie Jules Verne, Amiens, France. [55]Institute of Human Genetics, University Medical Center Hamburg-Eppendorf, Hamburg, Germany. [56]UF de Génétique Clinique, CHU de Reims, Reims, France. [57]Laboratoire de Génétique Moléculaire et Génomique, FHU GenOMedS, CHU Rennes, Rennes, France. [58]IGDR (Institut de Génétique et Développement de Rennes)-UMR 6290, Université Rennes, CNRS, INSERM, Rennes, France. [59]Département de Pathologie, Institut Universitaire du Cancer Toulouse - Oncopole, Toulouse, France. [60]Service de Génétique Médicale, Institut de Génétique Médicale d'Alsace (IGMA), CHU Strasbourg, Strasbourg, France. [61]Centre de Référence Anomalies du Développement et Syndromes Malformatifs, Université Bourgogne Europe, CHU Dijon Bourgogne, Inserm, CTM UMR1231, équipe GAD, FHU TRANSLAD, Centre de génétique, Centre de référence Déficiences Intellectuelles de Causes Rares et Centre de référence GénoPsy, Dijon, France. [62]Service de Génétique Clinique, Centre de Référence 'Anomalies du Développement et Syndromes Malformatifs' de l'Inter-région Ouest, FHU GenOMedS, CHU Rennes Hôpital Sud, Rennes, France. [63]Laboratoire de Biochimie Site Bicêtre, Faculté de Pharmacie, Hôpitaux Universitaires Paris-Saclay, Centre de référence des Maladies Mitochondriales, Filière Filnemu, Paris, France. [64]Laboratoire de Génétique Chromosomique, CHU de Montpellier, Montpellier, France. [65]Centre de Génétique, Université Bourgogne Europe, CHU Dijon Bourgogne, Centre de Référence maladies rares 'Anomalies du Développement et syndromes malformatifs,' FHU-TRANSLAD, Dijon, France. [66]Neuropediatric Department, University Hospital Centre Toulouse, Toulouse, France. [67]Kids Neuroscience Centre, Kids Research Institute, Sydney, New South Wales, Australia. [68]TY Nelson Department of Neurology and Neurosurgery, The Children's Hospital at Westmead, Sydney Children's Hospitals Network, Sydney, New South Wales, Australia. [69]Specialty of Child and Adolescent Health, University of Sydney, Sydney, New South Wales, Australia. [70]Service de Pédiatrie, Centre Hospitalier Intercommunal de Créteil, Créteil, France. [71]Department of Pediatric Neurology, Angers University Hospital, Angers, France. [72]Institute of Medical Genetics and Applied Genomics, University of Tübingen, Tübingen, Germany. [73]Department of Pediatrics, University Hospital, Division of Neuropediatrics and Social Pediatrics, Rheinisch-Westfälische Technische Hochschule Aachen, Aachen, Germany. [74]Département de Neurologie Pédiatrique, Assistance Publique - Hôpitaux de Paris (APHP) Sorbonne Université, Fédération Hospitalo-Universitaire I2-D2, Hôpital Armand Trousseau-La Roche Guyon, Paris, France. [75]Service de Neuropédiatrie, Hôpital Jean-Verdier, Bondy, France. [76]Consultation de génétique, CCMR ANDDI rare, Centre Hospitalier d'Alençon, Alençon, France. [77]Department of Paediatrics and Child Health, University of Otago, Wellington, New Zealand. [78]Département de Génétique Médicale, Hôpital Henri Mondor, Assistance Publique des Hôpitaux de Paris, Créteil, France. [79]Service de Pédiatrie, Nantes Université, CHU de Nantes, Service de Pédiatrie, Nantes, France. [80]Laboratoire de Génétique, CHU de Reims, Reims, France. [81]INSERM U1266 (Krebs team), Institute of Psychiatry and Neuroscience of Paris (IPNP),Université Paris Cité, Paris, France. [82]Département de génétique, Assistance Publique - Hôpitaux de Paris (APHP), Hôpital Robert-Debré, Paris, France. [83]MVZ Klinikum Oldenburg, Oldenburg, Germany. [84]Harvard Medical School, Boston, MA, USA. [85]Division of Genetics and Genomics, Boston Children's Hospital, Harvard Medical School, Boston, MA, USA. [86]Center for Genomic Medicine, Massachusetts General Hospital, Harvard Medical School, Boston, MA, USA. [87]Service de Génétique, CHU (Centre Hospitalier Universitaire) de La Réunion, Saint-Denis, France. [88]Health Technologies Institute, Inserm, Paris, France. [89]Department of Neurosciences, University of California San Diego, La Jolla, CA, USA. [90]Rady Children's Institute for Genomic Medicine, San Diego, CA, USA. [91]Department of Genetics and Personalized Medicine, Institute of Clinical Medicine, University of Tartu, Tartu, Estonia. [92]Department of Clinical Genetics, Genetics and Personalized Medicine Clinic, Tartu University Hospital, Tartu, Estonia. [93]Department of Clinical Epileptology, Sleep Disorders and Functional Neurology in Children, University Hospital of Lyon (HCL), Lyon, France. [94]Lyon Neuroscience Research Center, Inserm U1028/CNRS UMR 5292, Lyon, France. [95]Laboratoire de Génétique Médicale, CHR Metz-Thionville, Hôpital Mercy, Metz, France. [96]Institute of Genetics and Cellular and Molecular Biology (IGBMC), INSERM-U964, CNRS-UMR7104, University of Strasbourg, Illkirch, France. [97]Medical Genetics Department, Brest University Hospital, Brest, France. [98]University of Brest, Inserm, EFS, UMR 1078, GGB, Brest, France. [99]Service de Génétique Clinique, CHRU Nancy, Vandoeuvre les Nancy, France. [100]Department of Pediatrics, Hôpital Nord-Ouest, Villefranche sur Saône, France. [101]INM, INSERM U 1298, Montpellier, France. [102]Clinical Genetic Unit, Reference Center for Congenital Anomalies, CHU Montpellier, University of Montpellier, Montpellier, France. [103]Division of Pediatric Neurology, Neuroscience Institute, Norcliff Center for Integrative Brain Research, Seattle Children's Hospital/University of Washington, Seattle, WA, USA. [104]Caribbean Reference Center for Neuromuscular Diseases, University Hospital Fort de France, Martinique, France. [105]Neurologie Pédiatrique et Génétique Médicale, CHU de Clermont-Ferrand, Clermont-Ferrand, France. [106]UMRS 1112, INSERM, Université de Strasbourg, Strasbourg, France. [107]Department of Medical Genetics, Centre of Medical Genetics, University and University Hospital Münster, Münster, Germany. [108]Institute of Human Genetics, Leipzig University Medical Center, Leipzig, Germany. [109]Department of Biomedical Informatics, Harvard Medical School, Boston, MA, USA. [110]Division of Genetics, Brigham and Women's Hospital, Harvard Medical School, Boston, MA, USA. [111]MVZ Humangenetik Bremen, Limbach Genetics, Bremen, Germany. [112]Laboratory of Molecular Genetics of Rare Diseases, Montpellier University Hospital, Montpellier, France. [113]Department of Medical Genetics, Gulhane Faculty of Medicine, University of Health Sciences, Ankara, Turkey. [114]Department of Systemic and Organ Diseases, Assistance Publique - Hôpitaux de Paris (APHP), Paris City University, Genomic Medicine, Cochin Hospital, Paris, France. [115]Service de Génétique Médicale, Centre Hospitalier Universitaire (CHU) de Bordeaux, Bordeaux, France. [116]INSERM U1211, University of Bordeaux, Bordeaux, France. [117]Service de Génétique Médicale, AP-HM, Marseille, France. [118]Inserm, MMG, U1251, Aix Marseille Université, Marseille, France. [119]Department of Pediatric Neurology and Reference Center for Rare Children Epilepsy and Tuberous Sclerosis, Hôpital Femme Mere Enfant, Centre Hospitalier Universitaire de Lyon, Lyon, France. [120]Department of Neurology, Epilepsy Center Ulm, University Hospital Ulm, Ulm, Germany. [121]Department Neurology, Section of Epileptology, Medical Faculty, University RWTH Aachen, Aachen, Germany. [122]Institute of Human Genetics, Medical Faculty and University Hospital Düsseldorf, Heinrich Heine University Düsseldorf, Düsseldorf, Germany. [123]Centre Hospitalier Universitaire de Nice, Inserm U1081, CNRS UMR7284, IRCAN, Université Côte d'Azur, Nice, France. [124]Clinical Genetics Department, Human Genetics and Genome Research Institute, National Research Centre, Cairo, Egypt. [125]Neuroscience Group, Murdoch Children's Research Institute, Parkville, Victoria, Australia. [126]Department of

Paediatrics, University of Melbourne, Royal Children's Hospital, Melbourne, Victoria, Australia. [127]Florey Institute and Murdoch Children's Research Institute, Melbourne, Victoria, Australia. [128]Big Data Institute, University of Oxford, Oxford, UK. [129]Centre for Human Genetics, University of Oxford, Oxford, UK. [130]Institut for Advanced Biosciences, Université Grenoble Alpes, INSERM U 1209, CNRS UMR 5309, Grenoble, France. [131]Department of Biostatistics and Reference Center for Developmental Abnormalities, Université Rouen Normandie, Normandie Université, Inserm U1245 and CHU Rouen, Rouen, France. [132]These authors contributed equally: Elsa Leitão, Amandine Santini, Benjamin Cogne. [133]These authors jointly supervised this work: Christel Depienne, Gaetan Lesca, Caroline Nava. ✉e-mail: christel.depienne@uk-essen.de; gaetan.lesca@chu-lyon.fr; caroline.nava@aphp.fr

## Methods

### Inclusion and ethics statement

This study was conducted in accordance with the ethical standards and regulations of all participating countries. Written informed consent was obtained for all patients from their parents or legal guardians, with an additional consent form for families agreeing to the publication of photographs. For genetic analyses, patient samples were pseudonymized at each participating center. Information on the patients' sex (but not gender) was extracted from clinical records. The promoters of this research study are Assistance Publique–Hôpitaux de Paris (AP–HP) for hospitals associated with the SeqOIA laboratory (project ID APHP241333) and Grenoble-Alpes University Hospital (CHU Grenoble-Alpes, research ID 19814188) for hospitals affiliated with the Auragen laboratory. Ethical approval was obtained from the University Hospital Essen (24-12010-BO) and the Comité Éthique et Scientifique pour les Recherches, les Études et les Évaluations dans le domaine de la Santé (CESREES; reference 21082803 Bis / 2038764). AP–HP has received an authorization from the Commission Nationale de l'Informatique et des Libertés (CNIL; reference HGTHGT/MFIMFI/AR2426865; request no. 924924336666) for data processing. Additional approvals were obtained from the ethics committee of CHU de Nantes (CCTIRS number 14.556) and from CPP Ouest V (File 06/15, Ref MESR DC 2017 2987; approval date 4 August 2015). For methylation analyses, DNA from patients and controls had been previously collected in a medical context for genetic testing, with written consent including authorization for research use of leftover material. Control samples consisted of participants without neurodevelopmental disorders, either unaffected relatives or persons tested presymptomatically for other conditions who were found not to carry pathogenic variants. DNA samples used for methylation profiling were stored within the genetics biobank of the CRBi, Rouen, France (collection DC 2008-711, authorization MCRBi/2024/02). The use of these samples was approved by the CERDE ethics committee of Rouen University Hospital (notification E2023-13). Researchers and clinicians from all contributing centers participated throughout the study, from design and implementation to data collection, analysis and manuscript preparation, and are listed as co-authors of this article.

### snRNA genes

Genes and pseudogenes corresponding to snRNAs were retrieved from Ensembl 113 BioMart[60,61] by filtering on gene type 'snRNA.' Of the 2,094 genes, we excluded the ones placed in contigs, scaffolds or patches, keeping 1,901 genomic regions located in identifiable chromosomes. Of these, 1,741 genes are annotated as spliceosomal snRNAs. We downloaded through the UCSC Table Browser[62]: (1) the coordinates of the ENCODE Registry of candidate cCREs in the human genome[26] and (2) the peaks of histone H3 acetylation of lysine 27 (H3K27ac) obtained for the H1 human embryonic stem cell line (H1-hESC)[27]. We annotated the 1,901 snRNAs by analyzing their genomic coordinates for overlaps with ENCODE cCREs and H3K27ac peaks. The *RNU* genes were further annotated for their hypermutability[25]. We kept for further analyses 200 putative functional snRNA genes that either overlap with ENCODE cCREs, are considered hypermutable or are approved by the HUGO Gene Nomenclature Committee (HGNC).

### Small RNA-seq data from embryonic brain tissues

We analyzed the expression level of the 1,901 snRNAs using small RNA-seq data generated from six human embryonic brain regions by the ENCODE Consortium[63]: diencephalon (GSE78292), temporal lobe (GSE78303), occipital lobe (GSE78298), frontal cortex (GSE78293), parietal lobe (GSE78299) and cerebellum (GSE78291). We downloaded 24 bigwig files from the ENCODE portal[26] containing signals for 'all reads' and 'unique reads,' for plus and minus strands, from the default anisogenic replicate. For each genomic coordinate, we calculated the maximum normalized RNA-seq signal obtained from the bigwig files for the corresponding gene strand, as well as the number of covered bases. For plotting, we considered only snRNA genes for which at least 50% of the expected number of bases for the specific snRNA type were covered in at least one tissue when considering 'all reads.' The expected number of bases for each snRNA type were 164 (U1), 191 (U2), 141 (U4), 116 (U5), 107 (U6), 63 (U7), 127 (U4ATAC), 126 (U6ATAC), 134 (U11), 150 (U12) and 40 (other snRNAs).

### Patient cohorts and variant filtering

We first investigated monoallelic de novo and biallelic variants in snRNA genes in patients who underwent genome sequencing as part of the diagnostic process in France through Plan France Médecine Génomique 2025 (PFMG2025)[28] on one of the two national clinical sequencing laboratories, SeqOIA (https://laboratoire-seqoia.fr/) and Auragen (https://www.auragen.fr/). Sequencing was performed in two subsets (Auragen and SeqOIA), with final inclusion dates in May and November 2025, respectively. gLEAVES, the system used for genome analysis on the SeqOIA platform, which is restricted to registered and accredited users, was used to investigate inheritance of variants and visualize bam files on Integrative Genomics Viewer (IGV). Because bioinformatics pipelines were run independently in each laboratory and differed slightly, each dataset was analyzed both separately and combined. The combined cohort included 34,329 patients with rare disorders (20,690 from SeqOIA and 13,639 from Auragen). Characteristics of the combined PFMG cohort and separated subcohorts and details of the filters appear in Extended Data Fig. 1 and Supplementary Fig. 4. Variants from 200 snRNA genes were extracted initially using the criteria: de novo or biallelic inheritance, and present in fewer than three homozygotes in gnomAD v.3 and fewer than five homozygotes in internal databases. Biallelic variants were defined as: (1) homozygous or hemizygous variants; (2) two heterozygous variants inherited from different parents for trios; (3) one inherited and one variant absent from the sequenced parent for duos or (4) one inherited and one de novo variant, with a *trans* configuration verified manually in IGV. Only variants with coverage at least ten in genes with median VAF ≥ 0.3 with less than 50% overlap with problematic regions in 'Genome in a Bottle' were considered for further analysis. Additional filtering for gene enrichment analyses included variants with gnomAD AC < 100 ('rare variants') and no gnomAD flags (for example, variants flagged as 'LCR'). Two-sided Fisher's test was used to test statistical enrichment in unsolved versus solved and in NDD versus non-NDD cases. Genes were kept for these analyses when the minimum number of patients carrying variants was sufficient to reach statistical significance ($P < 0.05$) in the cohort before multiple testing (unsolved versus solved: ten [PFMG, SeqOIA, Auragen]; NDD versus non-NDD: nine [PFMG, SeqOIA] or ten [Auragen]).

After the initial filtering, the criteria to select *RNU2-2* variants in the PFMG cohort were relaxed to broaden the inclusion of potentially pathogenic variants, restricting only on the homozygote frequency (<3 in gnomAD_v3 and <5 in internal databases) while removing restrictions on heterozygote frequency. We examined all cases with biallelic variants manually, and transitioned to the AoU database, which encompasses a larger cohort of genomes from healthy people, thereby providing greater power to assess population-level variation. Refined criteria were: AC < 50 for de novo variants and AC < 200 for biallelic variants and no homozygotes in AoU. Cases with de novo or P/LP variants were screened for a second variant in *trans*, revealing potentially pathogenic or modifier alleles with slightly higher population frequencies in a few families. Variant nomenclature was checked systematically with Variant Validator (https://variantvalidator.org/service/validate/).

To identify additional cases with *RNU2*-2 variants, we performed sequencing of *RNU2-2* and/or reanalyzed genome data in a cohort of 5,456 patients with NDDs (Extended Data Fig. 1 and Supplementary Table 7). Furthermore, cases with biallelic variants present in 24,958 genomes were identified using seqr[29]. This large international collaborative effort identified 34 additional cases: 13 patients

with monoallelic variants (n.4G>A, *n* = 9; n.35A>G, *n* = 3; n.38A>G, *n* = 1) and 21 families with biallelic variants, originating from Germany (*n* = 9), France (*n* = 8), the US (*n* = 6), New Zealand (*n* = 3), Egypt (*n* = 3), Australia (*n* = 2), Denmark (*n* = 1), Turkey (*n* = 1) and Brazil (*n* = 1). None of these patients had been published previously, and we carefully verified the absence of duplicates among people carrying the same variant(s) by cross-checking year of birth and initials.

## Variant classification

We classified variants according to the American College of Medical Genetics and Genomics/Association for Molecular Pathology criteria using the recommendations of Ellingford et al.[64]. For de novo variants, PS2 was applied but downgraded in light of the hypermutable nature of snRNA genes: PS2_Supporting was applied for one de novo, PS2_Moderate when the same de novo variant was found in two to ten patients, and PS2 when the same de novo variant was found in more than ten patients (for n.4G>A and n.35A>G). PM2_Supporting was applied to variants absent or very rare in gnomAD v.3 (allele count <10) and AoU (allele count <50). PM1 was applied to variants located in functional domains U2/U6 helix II, SLI and BSL (n.1–n.45) and Sm binding site (n.98–n.107) and PM1_Supporting to variants in n.47-n.66 (SLIIa), n.68-n.84 (SLIIb), n.112–n.144 (SLIII) and n.147–n.184 (SLIV). PS4_Supporting was applied for variants present in two unrelated patients, PS4_Moderate when observed in three to four unrelated patients, and PS4 when present in five or more unrelated patients when the variant is rare (that is, PM2_sup is met). PS4 was downgraded for variants with allele count comprised between 11 and 50 in gnomAD v.3 and 51 to 250 in AoU (PS4_supp when observed in three to four unrelated patients, and PS4_Moderate when present in five or more unrelated patients). This criterion was not applied for more frequent variants. PP1 was applied if a variant cosegregated with disease in two or three affected family members, PP1_Moderate was applied when cosegregation was observed in five family members (affected or unaffected) or across several unrelated families. PM3 was assigned to variants detected in *trans* with a likely pathogenic or pathogenic variant.

We also annotated variants with MobiDeep available on Mobidetails website (https://mobidetails.chu-montpellier.fr)[65]. MobiDeep is a metascore for noncoding variants based on a multilayer perceptron using five features: REMM v.0.4, CADD 1.7, GPN-MSA and two conservation scores (Cactus 241-way vertebrates and PhyloP primates) to capture different evolutionary depths. Variants were stratified according to MobiDeep thresholds: <0.6 (neutral), >0.6 (probably deleterious) and >0.9684 (high-confidence deleterious).

## Sanger sequencing

Sanger targeted sequencing was performed to screen for variants in *RNU2-2* and/or to perform segregation analysis. PCR amplification of *RNU2-2* was performed using the HotStarTaq Master Mix Kit (Qiagen, cat. no. 203445) with the following primers: F: 5′- CAAACACGCGT CATTCAACACAC-3′; R: 5′- ATAACTGGTTGGAAGATGGGAAGG-3′ (designed using primer3), according to the manufacturer's instructions. Forward and reverse sequencing reactions were performed using the BrilliantDye Terminator v.1.1 Cycle Sequencing Kit (Nimagen; cat. no. BRD1-1000) or the BigDye Terminator v.3.1 Sequencing Kit (Life Technologies, cat. no. 4337457). ExoSAP-Purified sequencing products (ExoSAP-IT, Applied Biosystem, cat. no. 78205) were run on Pop-7 polymer (Life Technologies, cat. no. 4335615) using an ABI 3730 or 3730XL automated sequencer (Applied Biosystems). Sequences were analyzed using Geneious Prime 2019 (Biomatters Ltd.) or Seqscape v.2.6 software (Applied Biosystems).

## Clinical analyses

Clinical data were collected retrospectively from the referring physician using an anonymized Excel sheet, as published previously[18]. We collected categorical data on 66 clinical features from 112 patients.

For clustering and statistical comparisons, we used 60 features from 87 index patients, except in the analysis restricted to patients carrying biallelic variants, where 59 features from 56 patients were included since the remaining features showed no variation among patients. Data were converted to a 0–1 scale, with 0 representing a more favorable phenotype presentation and 1 a more severe phenotype. Hierarchical clustering was performed using pheatmap R package, performing *Z* score scaling for each row (across different patients), and ward.D2 clustering method keeping missing values. PCA was generated after replacing missing data by 0 and performing variable scaling. Microcephaly was defined as head circumference measurements less than the third percentile. We used charts established by Fenton et al.[66] to calculate head circumference percentile at birth and define congenital microcephaly. Fisher's tests (two-sided; 2 × 2, 2 × 3, 2 × 4 or 2 × 5 contingency tables) adjusted for multiple comparisons using Bonferroni correction were used to compare clinical features from patients with different variant types (n.4G>A versus n.35A>G: 39 (index only) or 37 (including siblings) clinical features; de novo versus biallelic variants: 48 (index only) or 50 (including siblings) clinical features). Clinical features were kept for the statistical analysis when the minimum number of patients with gathered information was sufficient to reach statistical significance (*P* < 0.05) before multiple testing.

## Prediction of variant impact

Structural analysis of variants and corresponding figures were performed using the PyMol v.3.0.0 visualization software[67] on published coordinates of the human U2 snRNP structures: PDB 7Q3L ref. 30 for the BSL structure and 5XJC ref. 31 for the representation of the rest.

## RNA sequencing

RNA-seq experiments were conducted following a similar protocol and workflow previously described for *RNU4-2*[18]. Briefly, PBMCs were isolated from 2–4 ml of EDTA blood and cultured in lymphocyte-stimulating medium for 48–72 h. RNA was extracted and stranded RNA-seq libraries prepared from 100 ng total RNA using the SureSelect XT-HS2 kit (Human All Exon V8 capture probes; cat. no. G9774C), followed by sequencing on an Illumina NextSeq 550 to obtain ~25–30 million paired-end reads per sample. Reads were aligned to GRCh38 with STAR v.2.7.11a. Quality control was performed with FastQC v.0.11.3 and Fastp v.0.23.4. Cell type composition was estimated using CIBERSORTx v.1.0 with the LM22 signature matrix. For splicing analysis, we focused on 19 people carrying either dominant (n.4G>A, *n* = 5; n.35A>G, *n* = 5) or biallelic (*n* = 9) *RNU2-2* variants that we compared to 49 controls (probands without *RNU2-2* variants). rMATS-turbo v.4.3.0 was used to detect alternative splicing events with the following parameters: -t paired –anchorLength 1 –libType fr-firststrand – novelSS –variable-read-length –allow-clipping. rMATS outputs were filtered for mean coverage >10 and ΔPSI > 0.05. For each splicing event, we fitted a linear regression model adjusting for age, sex and estimated cell composition using ordinary least squares with statsmodels (v.0.14.5) in Python (v.3.12.2). *P* values were obtained for association with affected status; significant splicing events were defined as those with *P* value < 0.01, retaining only the most significant event per gene. To visualize results, the linear regression model was trained on the 49 control samples and applied to both controls and affected patients; residual PSI values were then used for PCA using sklearn (v.1.7.2) and clustermaps using seaborn (v.0.13.2) for the splicing categories: SE, 5′SS, 3′SS and RI. Sashimi plot was performed with rmats2sashimi.

## DNA methylation study

DNA methylation analysis was performed as described previously[18]. Genomic DNA was extracted from whole blood and bisulfite converted. DNA methylation profiles were obtained using Infinium MethylationEPIC v.2.0 BeadChips (Illumina, cat. no. 20087708) following the manufacturer's protocol. Patients and controls were balanced across

50 arrays and array rows to reduce technical bias and matched for age at sampling. Arrays were processed at the ASGARD-Rouen genomic platform (Rouen University Hospital, University of Rouen) on an Illumina NextSeq 550 scanner. Raw IDAT files were normalized using Meffil R package, including all 50 arrays to estimate variability within and across arrays. Functional normalization was performed with random effect adjustment for array and sentrix row and fixed effect adjustment for the first two PCs, generating $\beta$-values.

An epigenome-wide association study was conducted using normalized $\beta$ values from controls and patients with P or LP dominant or biallelic *RNU2-2* variants. Multiple linear regression was applied per CpG to assess methylation differences, adjusting for age, sex and predicted blood cell composition. To account for potential heterogeneity, two variant-specific coefficients (dominant n.35A>G and biallelic recessive) were added alongside the main case–control effect, with n.4G>A as the reference. Regression coefficients and *P* values were extracted for all three variables. CpGs were considered significant if $P < 1 \times 10^{-5}$ and absolute effect size >0.05 for any coefficient. Significant CpGs were classified based on association patterns. Probes with only a main differential effect were labeled 'common to all variants.' Probes significant only for n.35A>G or biallelic coefficients were labeled 'specific to n.35A>G' or 'specific to biallelic recessive carriers,' respectively. Probes significant for both main and variant-specific coefficients corresponded to n.4G>A-specific probes after visual confirmation of effect directions.

Adjusted methylation levels were visualized using PCA and heatmaps (pheatmap, Euclidean distance, Ward aggregation method). Baseline methylation models were fitted on controls, correcting normalized $\beta$ value for age, sex and blood composition. CpGs were grouped by association patterns (common, n.35A>G-specific, biallelic-specific, n.4G>A-specific) and mean adjusted methylation per sample was calculated. Raincloud plots were generated (ggplot2, ggdist) combining half-violin distributions, boxplots and jittered points colored by group. Pairwise comparisons were performed using two-sided Wilcoxon rank-sum tests (ggpubr), with displaying significant *P* values only.

Predictive performance of the CpG signature was assessed using a multiclass linear kernel SVM (e1071) with four outcomes: Control, Dominant n.4G>A, Dominant n.35A>G, Recessive biallelic. Three-block cross-validation was applied within each variant group, training on two blocks and testing on the third, ensuring each sample was predicted once. For each cross-validation iteration, a new differential methylation analysis was performed and the SVM classifier was trained on the resulting CpG signature. Predicted labels and class probabilities were obtained, and sensitivity and specificity per class were calculated with 95% confidence intervals (Wilson method). Probabilities were averaged and visualized to illustrate class separation and model confidence.

#### Reporting summary
Further information on research design is available in the Nature Portfolio Reporting Summary linked to this article.

### Data availability
Variant details have been submitted to ClinVar (SUB15855213). RNA-seq and methylation data have been deposited in the European Genome-phenome Archive (EGA, http://www.ebi.ac.uk/ega), hosted by the EBI. RNA-seq data are available under the Study Accession Number EGAS50000001410. Methylation data are accessible under the study accession EGAS00001008070. Both are subject to a Data Processing Agreement, and access requests will be reviewed by a Data Access Committee to ensure compliance with ethical and legal standards. Due to ethical considerations, individual genome data cannot be made publicly available. Controlled access is required to safeguard participant privacy and to comply with data protection regulations, including the GDPR in Europe. Access to genome data from the PFMG2025 cohort is governed by French data protection laws and is only possible through the Collecteur Analyseur de Données (CAD). More details can be found on the PFMG2025 website at https://pfmg2025.fr/le-plan/collecteur-analyseur-de-donnees-cad/. The coordinates of the ENCODE Registry of cCREs in the human genome[26] and bigwig files concerning the peaks of histone H3 acetylation of lysine 27 (H3K27ac) obtained for the H1 human embryonic stem cell line (H1-hESC)[27] were downloaded through the UCSC Table Browser[62]. Bigwig files concerning small RNA-seq data from six human embryonic brain regions were downloaded from the ENCODE portal[68] (https://www.encodeproject.org/) with the following identifiers: ENCFF013RLG, ENCFF029RIV, ENCFF034QAV, ENCFF-197SSE, ENCFF221KEN, ENCFF343ZBS, ENCFF405QIN, ENCFF532SOY, ENCFF654ONK, ENCFF738LDD, ENCFF870FMA, ENCFF887TOS, ENCFF106ESQ, ENCFF222WBQ, ENCFF250WEA, ENCFF254UEQ, ENCFF319GRF, ENCFF425WUZ, ENCFF443ONL, ENCFF820JTT, ENCFF897IWP, ENCFF915WAC, ENCFF946YVE and ENCFF965GHD.

### Code availability
No custom code was developed for this study. All analyses were performed using existing, publicly available pipelines and software tools listed in the Methods and Reporting Summary.

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

### Acknowledgements
We are deeply grateful to the patients and their families for their participation in this study. Patients included in this study were diagnosed through several studies. We thank S. Mazoyer for critical reading of the preprint, which led to text edits in the final version of this article. Patients from the PFMG received funding from Plan France Médecine Génomique 2025 (PFMG2025). Some of the results were supported by AP–HP and the RNU-SPLICE project, financed by Health philantropic program of Mutuelles AXA dedicated to supporting innovative research projects in France (to C.N.). The structural interpretation was conducted by C. Charenton and M.A. as part of the ERC Starting Grant project 'SPLIFEM.' A. Santini received a grant from Region Normandie and GIRCI Nord Ouest (FHU-A2M2P). Sequencing and/or analysis of cases identified in Germany were supported by grants from the Deutsche Forschungsgemeinschaft (DFG), project number 458099954 (DE 2979/6-1) to C. Depienne and KU 1240/17-1 to K.K. Sequencing and analysis of cases identified by the Broad Institute of MIT and Harvard Center for Mendelian Genomics (A.O'D.-L., O.M., A.N., M.C.O'L., L.S.P., C.A.-T. and S.L.S.) were funded by the National Human Genome Research Institute (NHGRI) grants UM1HG008900 (with additional support from the National Eye Institute, and the National Heart, Lung and Blood Institute), U01HG011755, R21HG012397 and R01HG009141, and in part by the Chan Zuckerberg Initiative

Donor-Advised Fund at the Silicon Valley Community Foundation (funder https://doi.org/10.13039/100014989) grants 2019-199278, 2020-224274, 2022-316726 and 2022-309464 (https://doi.org/10.37921/236582yuakxy). The content is solely the responsibility of the authors and does not necessarily represent the official views of the funding agencies. Genetic analyses (cases from Gleeson laboratory) were supported by the Gabriella Miller Kids First Pediatric Research Program, funded by the Common Fund of the NIH Office of the Director, with sequencing performed at the Broad Institute Sequencing Center (U24HD090743). Genetic analyses of Australian cases were supported by the Genetic Basis of Epilepsy Research program (NHMRC grants GNT1172897, GNT2033247, GNT2006841) and by Victorian State Government Operational Infrastructure Support and the NHMRC IRIISS. M.F.B., M.S.H. and I.E.S. were supported by an Australian Medical Research Future Fund Genomics Health Futures Mission grant (2007707). This study uses resources generated by the ENCODE Consortium: (1) the ENCODE Data Analysis Center (ZLab at UMass Medical Center), H. Pratt, J. Moore, M. Purcaro and Z. Weng for providing ENCODE cCREs data; (2) Bernstein laboratory at the Broad Institute for the H3K27ac peaks in H1-hESC; (3) T. Gingeras laboratory at Cold Spring Harbor Laboratory for generating the small RNA-seq data from six human embryonic brain regions. N.W. is supported by a Wellcome Career Development Award (grant no. 305292/Z/23/Z) and a research prize from the Lister Institute. H.C.M. is funded by ALSAC of St. Jude Children's Research Hospital. O.M. was supported by the Hazem Ben-Gacem Tunisia Medical Fellowship Fund. K.Õ. was supported by Estonian Research Council grants PRG471 and PRG2040. Health Research Council of New Zealand and Curekids New Zealand supported the recruitment, phenotyping and sequencing of the New Zealand participants. C. Charbonnier is supported by the 2022 MESSIDORE program (project Inserm-MESSIDORE 19). M. Essid, G.B., S. Auvin, S. Baer, N. Chemaly, A.d.S.M., M.M., R.N., E.P., J. Toulouse, N.V. and G. Lesca are members of the EPICARE European Reference Network (ERN-EPICARE). A.K., F.J.K., D.W., L.F., K.Õ., N.C.B., I.S. and P.S.-W. are members of the European Reference Network for Developmental Anomalies and Intellectual Disability (ERN-ITHACA). Support for title page creation and format was provided by AuthorArranger—a tool developed at the National Cancer Institute.

## Author contributions

E. Leitão performed the in silico analyses of snRNA genes, burden tests, genotype–phenotype correlations and statistical analyses. G.L., C.N., V.B., P.M. and J. Thevenon contributed snRNA variant data from the PFMG cohort. B. Cogne and T.B. performed RNA sequencing and data analysis. A. Santini and C. Charbonnier carried out DNA methylation experiments and analyzed episignatures. M.A. and C. Charenton conducted the structural analyses. G.L., C.N., M. Essid and C.W.L. collected clinical information. C.N., G.L., C. Charenton, A. Santini and B. Cogne contributed to sections of the manuscript and C. Depienne drafted the article. C. Depienne, C.N. and G.L. supervised the study. P.M., V.B., K.J., N. Chatron, G.B., B.K., C. Mignot, P. Charles, T.B., R.P., J.-M.d.S.A., E.P.A.F., S. Sengupta, M.M., F.R., T.A., I.A., C.A., S. Arpin, C.A.-T., S. Auvin, S. Baer, N.B.-B., M. Bak, M. Barth, S. Baulac, N.B.-W., M.B., M.F.B., U.B., S. Bézieau, R.B., M. Biehler, T. B. Hammer, J. Bogoin, E.B., S. Boussion, C.B., A.B.-B., A.-L.B., A.B.-S., J. Buratti, T. Celse, P. Chambon, N. Chemaly, B. Chesneau, E.C., M.C., C. Colson, S.C., T. Courtin, I.C., A.-C.C., L.T.D., A.d.S.M., C.d.V.d.B.L., B. Demeer, A.-S.D.-P., P.D., S. DiTroia, M.D.-F., C. Dubourg, C. Dubucs, S. Ducreux, L.D., R.D., B. Durand, S.E.C., M. Elbracht, L.F., M. Faoucher, A.F., S.F., M. Fradin, P. Gaignard, B.G., A. Garde, J.G., D. Gill, A. Goldenberg, D. Grabli, C.G., S.G., P. Gueguen, A.-M.G., A. Guichet, T. B. Haack, N.H., M.G.H., S.H., T. Herget, B.H., D.H., J.H., M.H., T. Holling, C.H., B.I., A.J., L.J., N.J.-M., F.J.K., S.K., C.K., M.K., F.K., J.K., R.K., A.K., I.K., K.K., A. Labalme, J.-S.L., V.L., F.L.B., A.-S.L., M.L., E. Leguern, J.L., N.L., S.L., K.L., S.M.W.M., L.M.-H., J. Maraval, I.M., C. Mattausch, S. Mercier, O.M., G.M., J. Mortreux, A.M., R.N., S.N., V.N., A.N., L.N., M.N., F.N., M.C.O'L., S. Odent, N.M.O., V.O., S. Olivieri, K.Õ., L.S.P., E.P., O.P., L.P.-S., F.P., C. Philippe, A. Piton, M.P., C. Poirsier, A. Pouzet, C. Prouteau, S.Q.-R., M. Renaud, A.-C.R., M. Rio, C.R., F.R.-R., P.R., M. Rossi, A.R., V.R., M.R.-M., P.S.-V., A. Saunier, R.S., E. Sarrazin, C.S., E. Schaefer, C.S.-B., A. Schneider, I.S., V.B.S., S. Spranger, T.S., M.S., S.R.S., B.S.-B., S.L.S., F.S., M.T., D.T., J. Toulouse, H.T., S. Valence, S. Valleix, J.V.-G., L.V., D.V., N.V., A.V., A.W., J.W., Y.W., D.W., T.W., M.Y., T.Y., K.Z.-K., M.S.Z., A.Z., N.C.B., A. Lermine, G.N., J.G.G., L.G.S., M.S.H., I.E.S., N.W., A.O'D.-L., H.C.M. and P.B. contributed molecular and/or clinical data and provided intellectual input. All authors approved the final manuscript.

## Funding

## Competing interests

N.W. receives research funding from Novo Nordisk and Biomarin Pharmaceutical. L.T.D. receives research funding from Stoke Therapeutics. L.G.S. receives funding from the Health Research Council of New Zealand and Cure Kids New Zealand. She has served as a paid consultant of the Epilepsy Study Consortium for consulting work for Epygenix Therapeutics, Ovid Therapeutics, Stoke Therapeutics, Takeda Pharmaceuticals, UCB and Zogenix. L.G.S. has received research grants and consultancy fees from Zynerba Pharmaceuticals and has served on Takeda and Eisai Pharmaceuticals scientific advisory panels. I.E.S. has served on scientific advisory boards for CAMP4 Therapeutics, Longboard Pharmaceuticals, Mosaica Therapeutics, Takeda Pharmaceuticals and UCB; has received speaker honoraria from Akumentis, Biocodex, Chiesi, Stoke Therapeutics, UCB and Zuellig Pharma; has received funding for travel from Stoke Therapeutics and UCB; has served as an investigator for Anavex Life Sciences, Biohaven Ltd., Bright Minds Biosciences, Cerebral Therapeutics, Cerecin Inc., Cereval Therapeutics, Encoded Therapeutics, EpiMinder Inc., ES-Therapeutics, Longboard Pharmaceuticals, Marinus, Neuren Pharmaceuticals, Neurocrine BioSciences, Praxis Precision Medicines, Shanghai Zhimeng Biopharma, SK Life Science, Supernus Pharmaceuticals, Takeda Pharmaceuticals, UCB, Ultragenyx, Xenon Pharmaceuticals and Zogenix; has consulted for Biohaven Pharmaceuticals, Eisai, Epilepsy Consortium, Longboard Pharmaceuticals, Praxis, Stoke Therapeutics and UCB and is a executive Director of Bellberry Ltd. and a Director of the Australian Academy of Health and Medical Sciences. She may accrue future revenue on pending patent WO61/010176 (filed: 2008): Therapeutic Compound; has a patent for SCN1A testing held by Bionomics Inc and licensed to various diagnostic companies; has a patent molecular diagnostic/theranostic target for benign familial infantile epilepsy (BFIE) [PRRT2] 2011904493 and 2012900190 and PCT/AU2012/001321 (TECH ID:2012-009). The other authors declare no competing interests.

## Additional information

**Extended data** is available for this paper at https://doi.org/10.1038/s41588-026-02547-5.

**Correspondence and requests for materials** should be addressed to Christel Depienne, Gaetan Lesca or Caroline Nava.

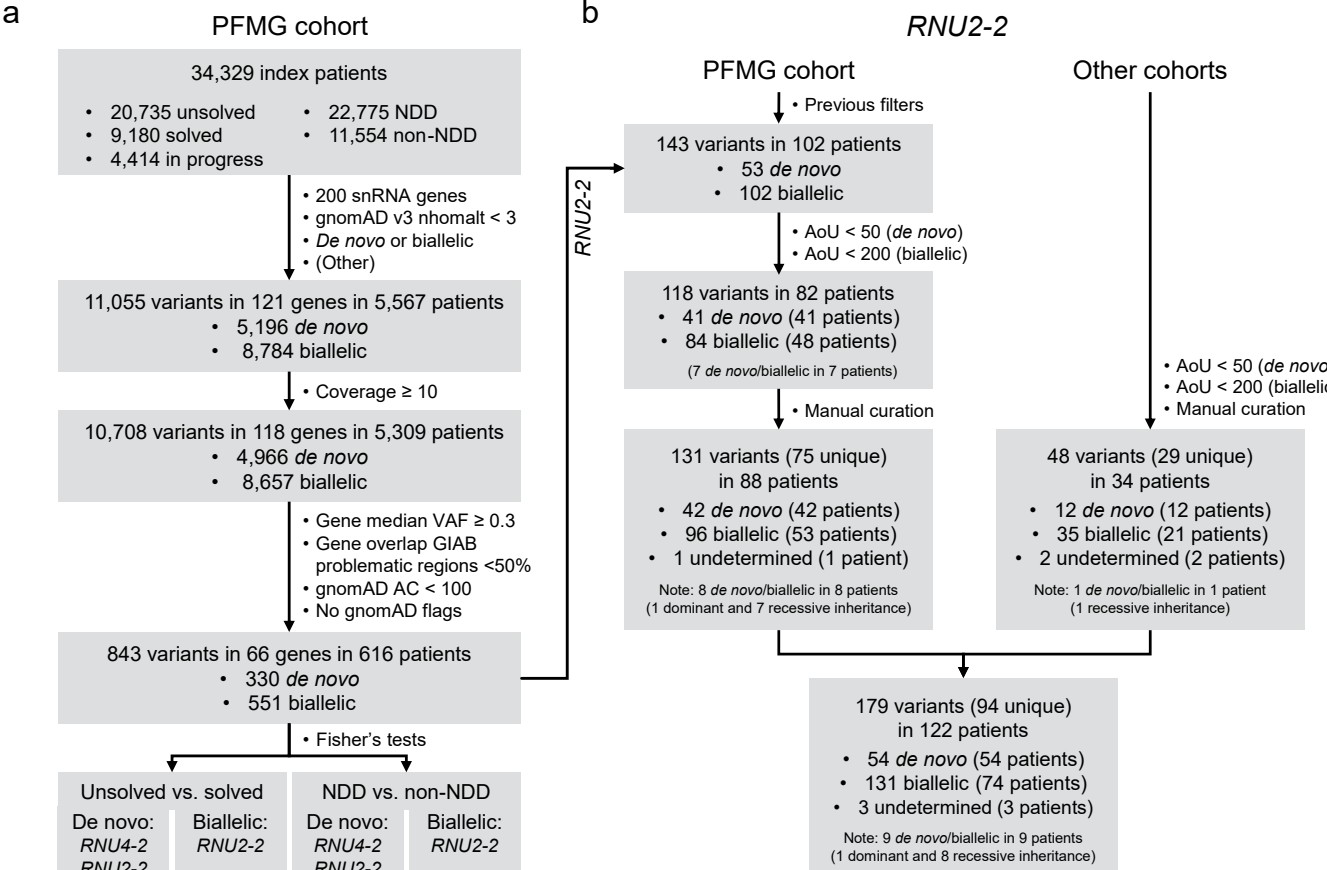

**Extended Data Fig. 1 | Details of the PFMG cohort and variant filtering strategy. a**, Number of patients and *de novo* or biallelic variants in snRNA genes the PFMG cohort. Supplementary Fig. 4 shows the same data in the SeqOIA and Auragen subcohorts separately. **b**, Number of patients with *de novo* or biallelic *RNU2-2* variants in the PFMG and other cohorts. Details of other cohorts appear in Supplementary Table 7. Note that the variant numbers include multiple occurrences of the same variants. The number of distinct ('unique') variants in *RNU2-2* appear in brackets. Variants *in cis* with *de novo* or biallelic variants are not counted.

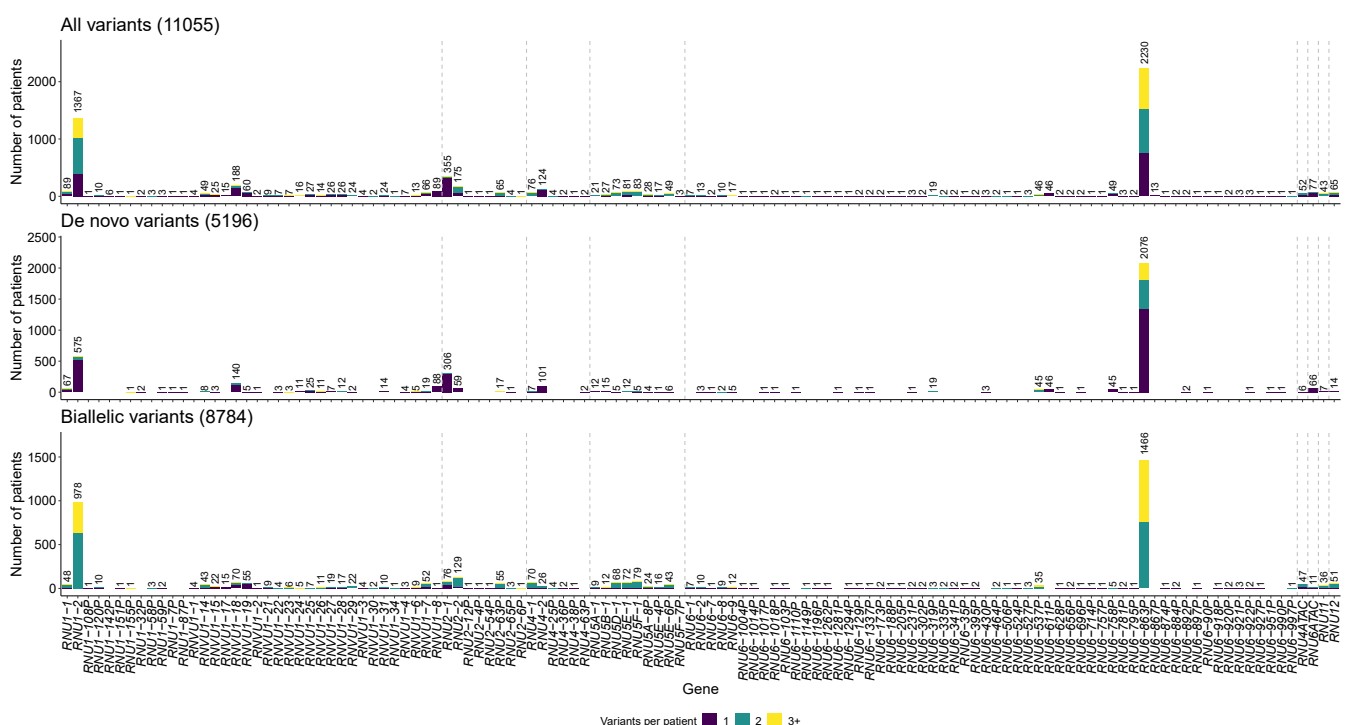

**Extended Data Fig. 2 | Distribution of variants across 200 putatively functional snRNA genes in the PFMG cohort.** Supplementary Figure 1 shows the same distribution in the SeqOIA and Auragen subcohorts separately. Variants were found in 121 of the 200 genes.

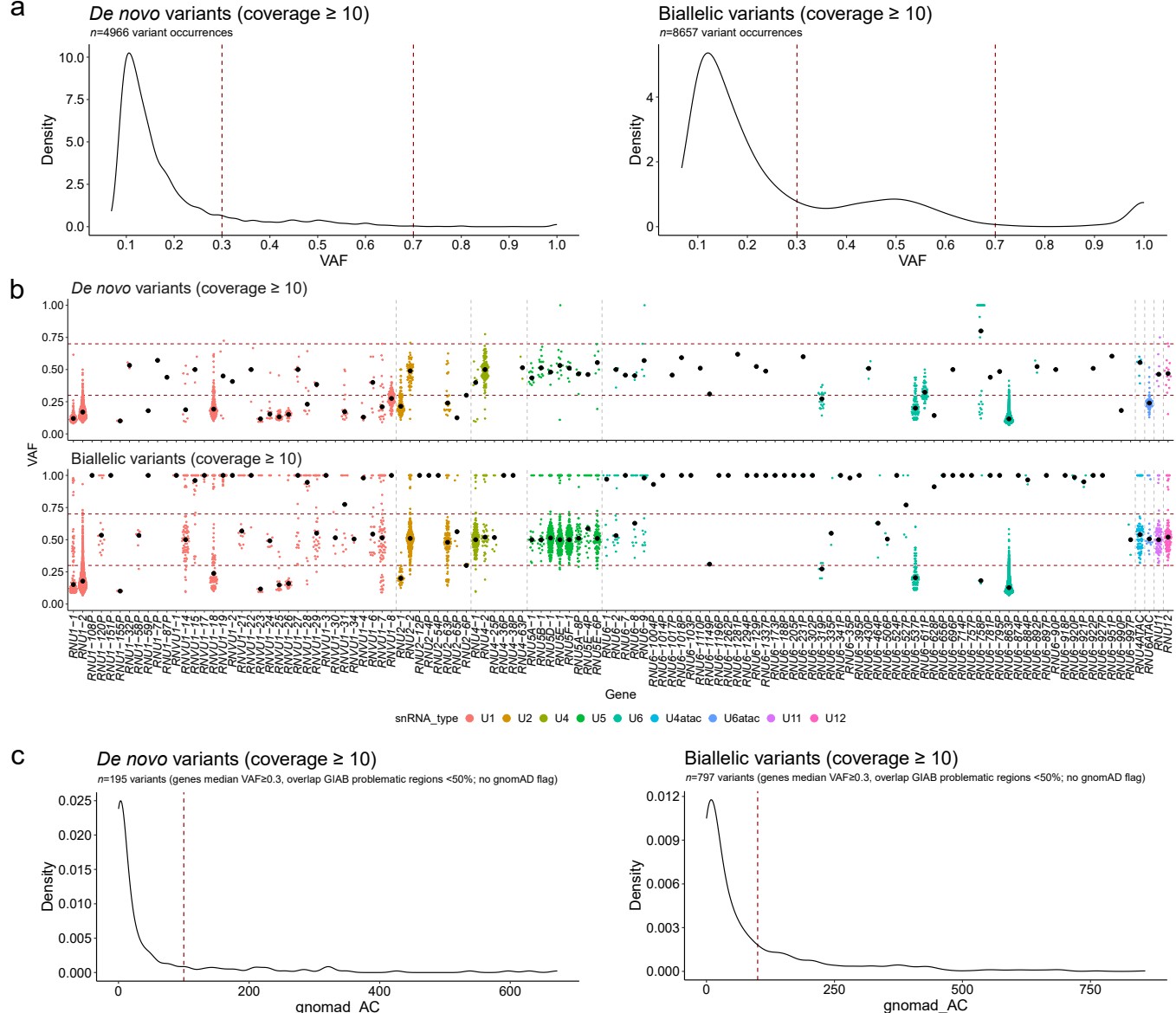

**Extended Data Fig. 3 | Variant allele fraction (VAF) distributions of snRNA variants in the aggregated PFMG cohort. a**, VAF distribution of *de novo* and biallelic variants in snRNA genes with coverage ≥ 10. **b**, Detailed VAF distribution for snRNA genes harboring variants in the PFMG cohort. Supplementary Fig. 2 shows the same data in the SeqOIA and Auragen subcohorts separately. **c**, gnomAD allele count (AC) distribution of *de novo* and biallelic variants in snRNA genes (with coverage ≥ 10) in the PFMG cohort. Supplementary Fig. 3 shows the same data in the SeqOIA and Auragen subcohorts separately.

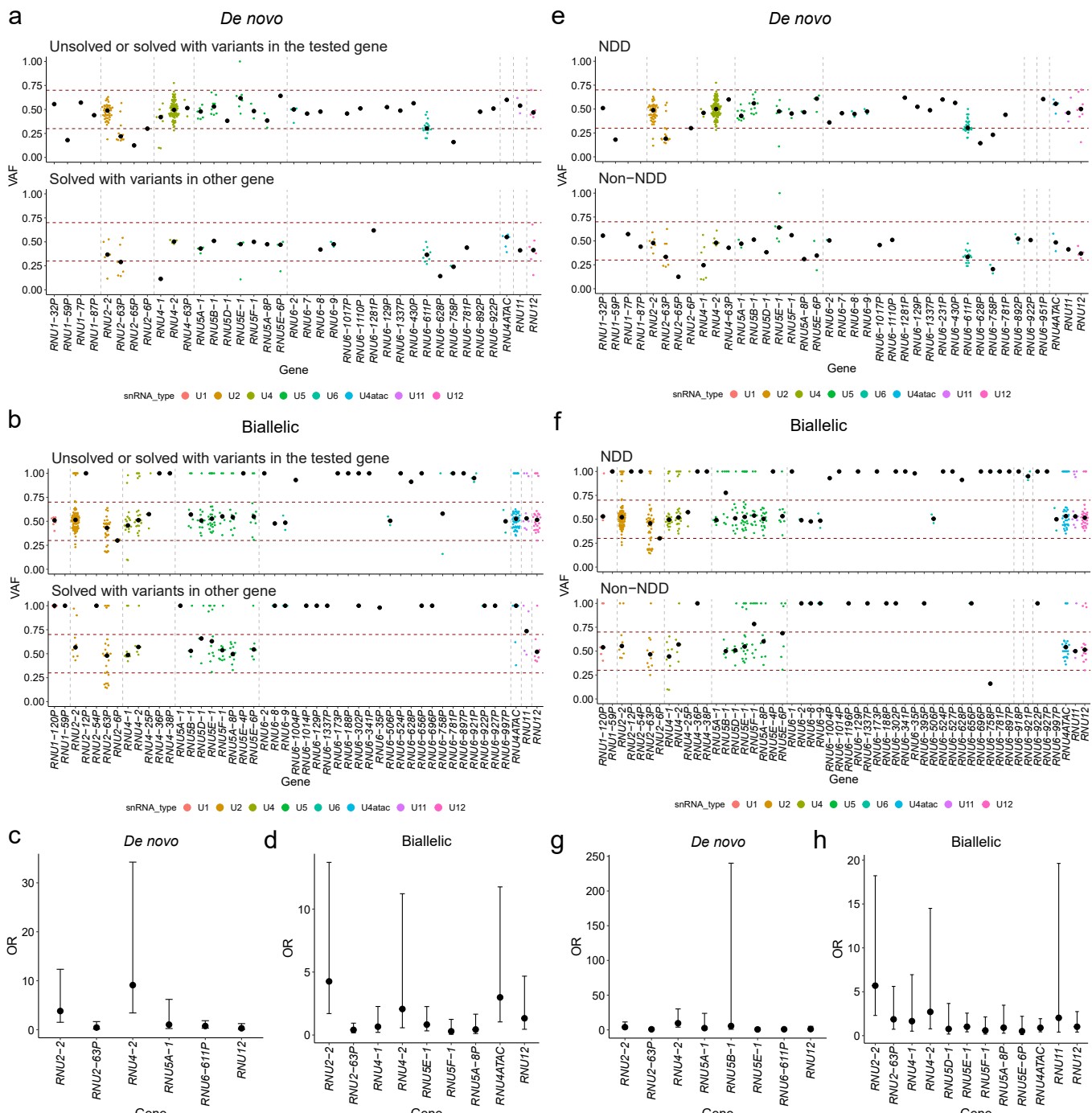

**Extended Data Fig. 4 | Identification of potential novel snRNA gene-disease associations in the PFMG cohort. a-d**, The cohort was divided into solved (*n* = 9,180) and unsolved (*n* = 20,735) cases for discovery analyses, with cases solved by variants in snRNAs with known disease association merged into the unsolved group. We compared the proportion of cases with rare variants (gnomAD allele count < 100) between solved and unsolved groups. **a,b**, Variant allele fraction (VAF) distribution for rare *de novo* variants (**a**) and rare biallelic variants (**b**). **c,d**, Odds ratio (OR; dot) and OR 95% confidence interval (error bars) of unsolved versus solved cases for genes in which at least 10 patients had *de novo* (**c**) or biallelic (**d**) variants (minimum number needed to reach statistical

significance in the cohort). **e-h**, The cohort was divided into NDD (*n* = 22,775) and non-NDD (*n* = 11,554) cases for discovery analyses. We compared the proportion of cases with rare variants (gnomAD allele count < 100) between NDD and non-NDD groups. **e,f**, Variant allele fraction (VAF) distribution for rare *de novo* variants (**e**) and rare biallelic variants (**f**). **g,h**, Odds ratio (OR; dot) and OR 95% confidence interval (error bars) of NDD versus non-NDD cases for genes in which at least 9 patients had *de novo* (**g**) or biallelic (**h**) variants (minimum number needed to reach statistical significance in the cohort). Two-sided Fisher's test *P*-values are shown in Fig. 2. Supplementary Figs. 5, 6 respectively show the same analyses performed separately in the SeqOIA and Auragen subcohorts.

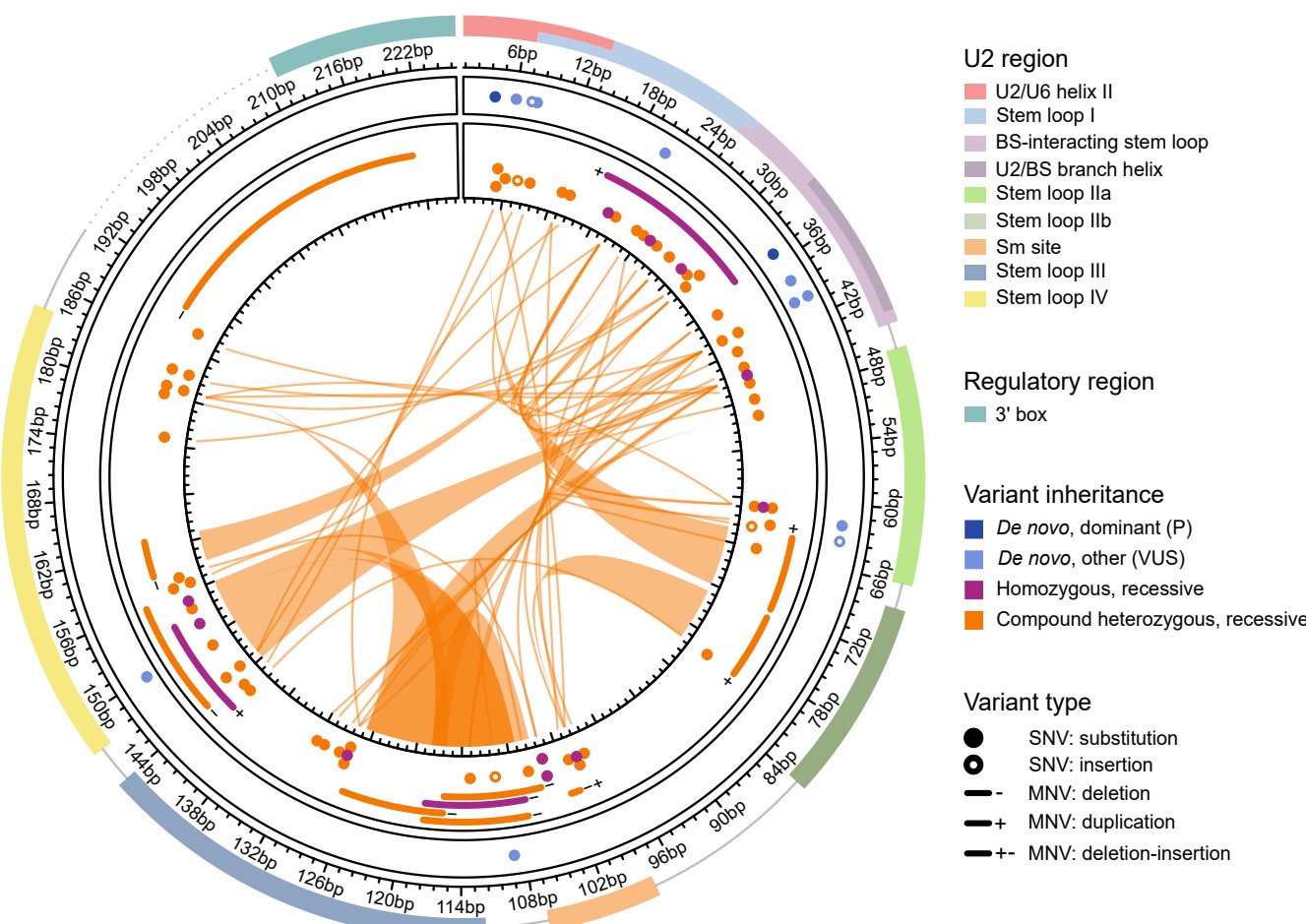

**Extended Data Fig. 5 | Circos plot depicting preferential associations of biallelic variants.** This scheme suggests that, in affected individuals with compound heterozygote variants, variants preferentially co-occur with one affecting the 5′ domain and the other the 3′ domain, suggesting domain-specific combinatorial effects. Variant coloring is the same as in Fig. 3a.

a

**n.107_118del (homozygous)**

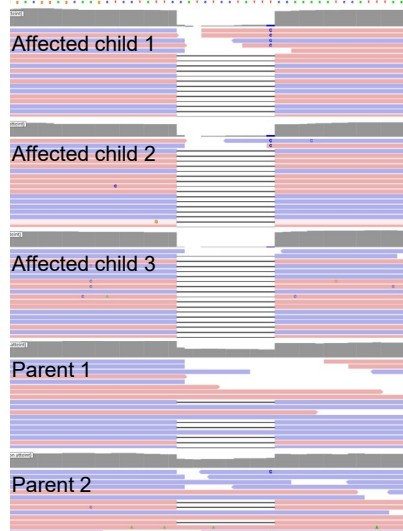

b

**n.149A>T (hemizygous); whole gene deletion**

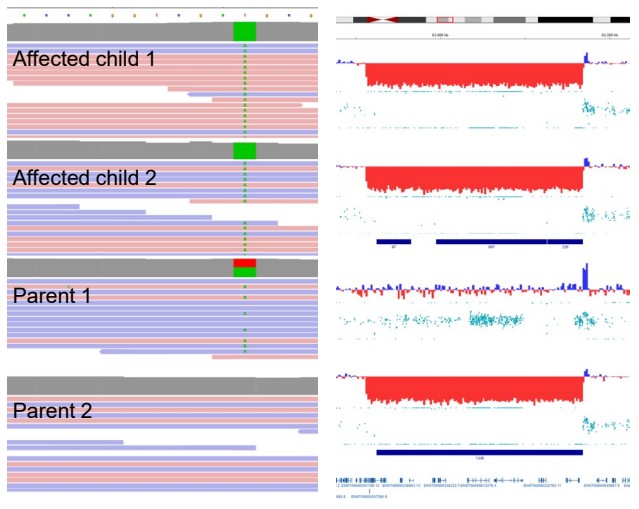

c

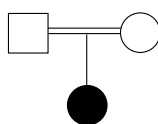

**n.142_153dup (homozygous)**

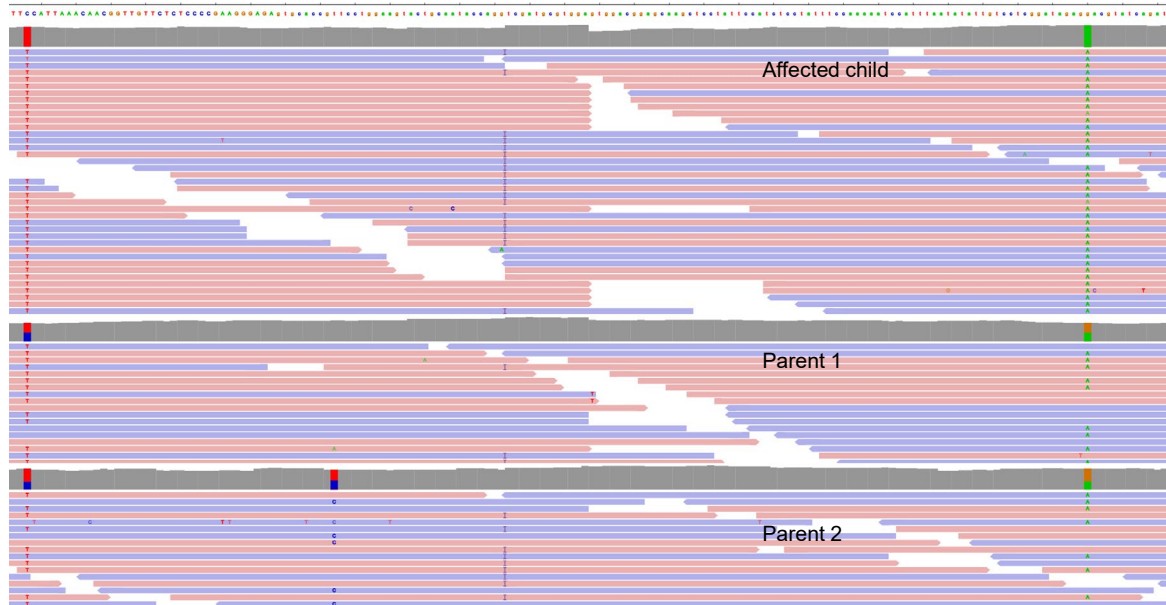

**Extended Data Fig. 6 | Examples of biallelic variants identified by genome sequencing in the PFMG cohort, with variant segregation consistent with recessive inheritance. a**, Integrative Genomics Viewer (IGV) screenshots of BAM file alignments showing segregation of the *RNU2-2* n.127_118del variant in the homozygous state in three affected sisters, with both parents carrying the variant in heterozygous state. **b**, Left: IGV screenshot displaying the hemizygous *RNU2-2*

n.149A>T variant in two affected siblings and their heterozygous mother. Right: Copy-number analysis of genome sequencing data revealing a 642-kb deletion (hg38: chr11:62,781,800-63,424,493) encompassing *RNU2-2* and 31 additional genes, present in the two affected siblings and their father. **c**, IGV screenshot showing the presence of the *RNU2-2* n.142_153dup variant in homozygous state in the affected child and in heterozygous state in both parents.

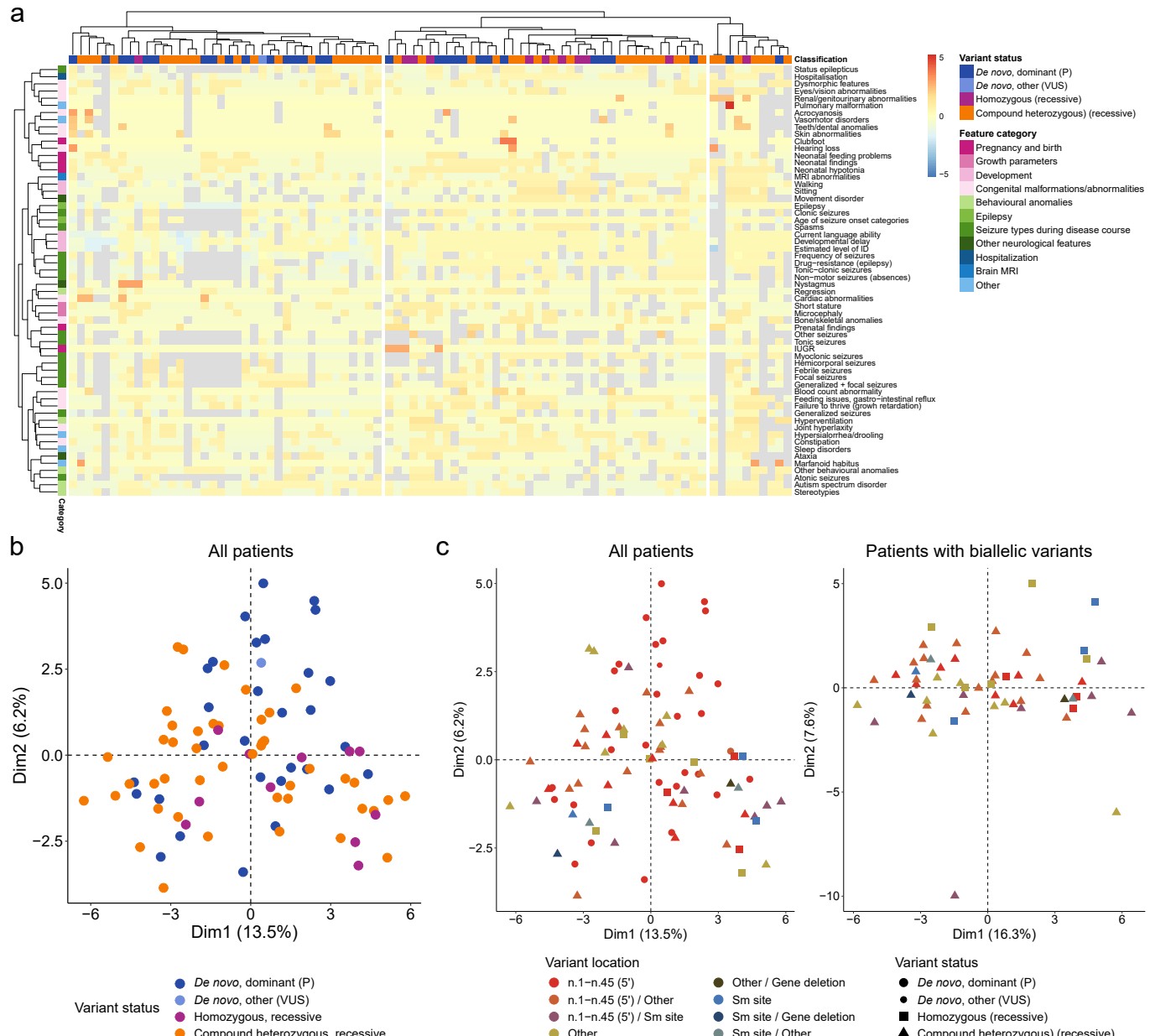

**Extended Data Fig. 7 | Patients with dominant and recessive *RNU2-2* NDDs have similar clinical features without obvious genotype-phenotype correlations.** **a**, Hierarchical clustering of the clinical features (*n* = 60, rows) of patients with *de novo* dominant or biallelic *RNU2-2* variants (*n* = 87, columns). Categorical data was converted to 0-1 scale, and values were *Z*-score scaled for each row. Blue-yellow-red scale depicts *Z*-scores. Lower values indicate a more favorable phenotype, while higher values represent a more severe phenotype. Missing values are shown in grey. Columns are colored based on the variant classification.

Dark blue: *de novo*, dominant (n.4G>A or n.35A>G; *n* = 30); light blue: *de novo* other (VUS; *n* = 1); orange: compound heterozygous, recessive (*n* = 45); purple: homozygous, recessive (*n* = 11). **b**, Principal component analysis of clinical features in *RNU2-2* variant carriers taking the variant status into account. Variant coloring is the same as in **a**. **c**, Principal component analysis of clinical features in *RNU2-2* variant carriers, considering variant type, location, and combinations. PCA on the left include all patients while PCA on the right include only patients with biallelic variants. Missing values were imputed as 0.

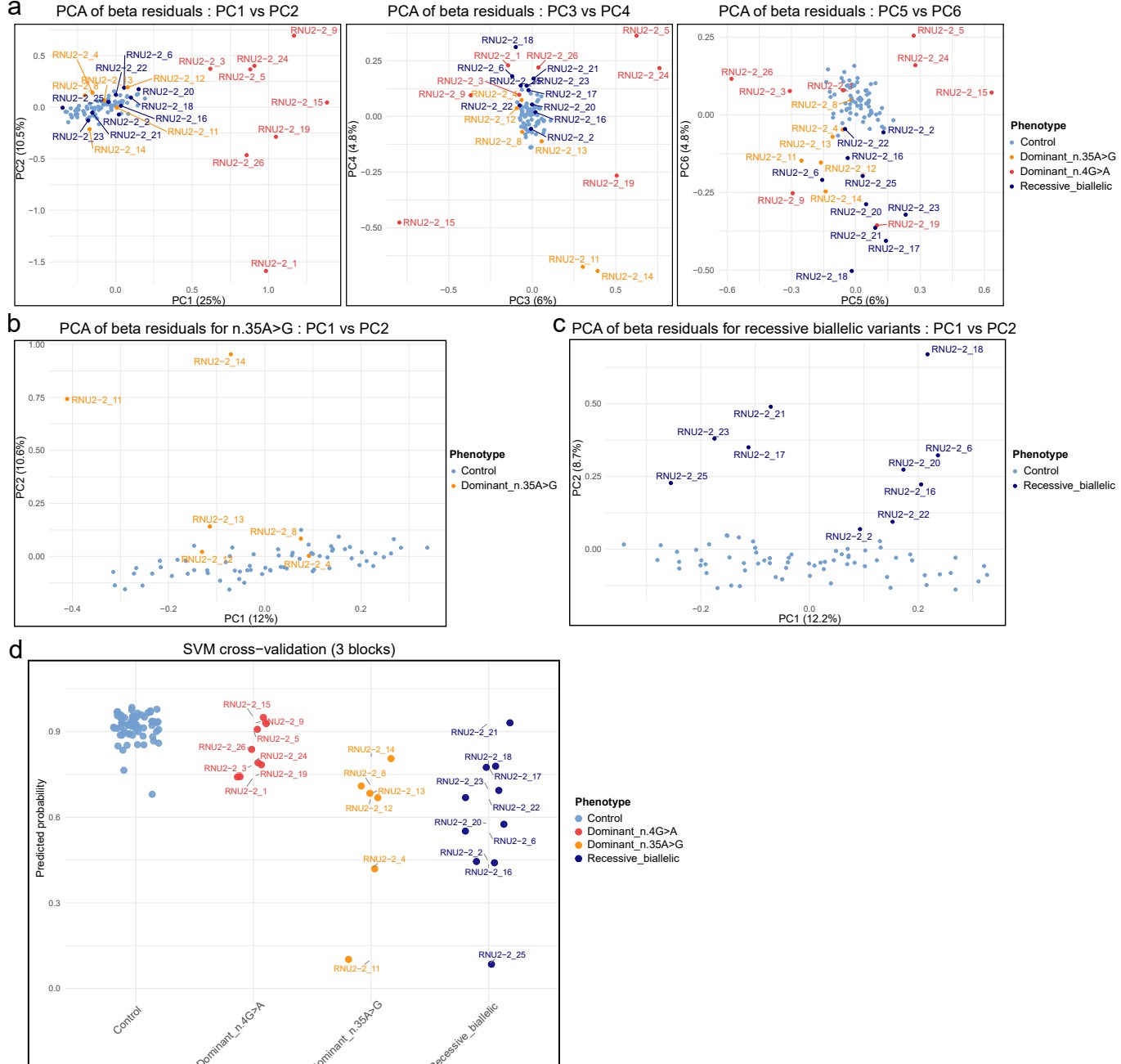

**Extended Data Fig. 8 | Principal component analysis and classification performance of the *RNU2-2* episignature.** In all panels, controls are shown in light blue, dominant n.4G>A carriers in red, dominant n.35A>G carriers in orange, and recessive biallelic cases in dark blue. **a**, PCA of adjusted methylation levels at differentially methylated positions (*n* = 201) up to component 6, after correction for expected baseline methylation level based on age at sampling, sex and estimated blood cell counts (*n* = 92 individuals in total) showing separation of dominant n.4G>A carriers from normal controls on axis 1, and the weaker separation between dominant n.35A>G carriers or recessive biallelic cases from normal controls along axis 6. The percentage of variance explained is provided for each component within the axis title. **b**, PCA of adjusted methylation levels at differentially methylated positions (*n* = 201), after correction for expected

baseline methylation level based on age at sampling, sex and estimated blood cell counts on the restriction to normal controls and carriers of the dominant n.35A>G variant along the first two principal components (*n* = 74 individuals in total). The percentage of variance explained is provided for each axis. **c**, PCA of adjusted methylation levels at differentially methylated positions (*n* = 201), after correction for expected baseline methylation level based on age at sampling, sex and estimated blood cell counts on the restriction to normal controls and carriers of recessive biallelic variants along the first two principal components (*n* = 78 individuals in total). The percentage of variance explained is provided for each axis. **d**, Predicted probabilities from a three-block cross-validation using a four-class SVM classifier (control, dominant n.4G>A, dominant n.35A>G, recessive biallelic).

# Reporting Summary

## Statistics

For all statistical analyses, confirm that the following items are present in the figure legend, table legend, main text, or Methods section.

| n/a | Confirmed | |
|---|---|---|
| ☐ | ☒ | The exact sample size ($n$) for each experimental group/condition, given as a discrete number and unit of measurement |
| ☒ | ☐ | A statement on whether measurements were taken from distinct samples or whether the same sample was measured repeatedly |
| ☐ | ☒ | The statistical test(s) used AND whether they are one- or two-sided<br>*Only common tests should be described solely by name; describe more complex techniques in the Methods section.* |
| ☐ | ☒ | A description of all covariates tested |
| ☐ | ☒ | A description of any assumptions or corrections, such as tests of normality and adjustment for multiple comparisons |
| ☐ | ☒ | A full description of the statistical parameters including central tendency (e.g. means) or other basic estimates (e.g. regression coefficient) AND variation (e.g. standard deviation) or associated estimates of uncertainty (e.g. confidence intervals) |
| ☐ | ☒ | For null hypothesis testing, the test statistic (e.g. $F$, $t$, $r$) with confidence intervals, effect sizes, degrees of freedom and $P$ value noted<br>*Give P values as exact values whenever suitable.* |
| ☒ | ☐ | For Bayesian analysis, information on the choice of priors and Markov chain Monte Carlo settings |
| ☒ | ☐ | For hierarchical and complex designs, identification of the appropriate level for tests and full reporting of outcomes |
| ☐ | ☒ | Estimates of effect sizes (e.g. Cohen's $d$, Pearson's $r$), indicating how they were calculated |

*Our web collection on statistics for biologists contains articles on many of the points above.*

## Software and code

Policy information about availability of computer code

| | |
|---|---|
| Data collection | Variants in genes encoding snRNA in the PFMG were accessed using custom scripts. gLEAVES, the system used for genome analysis on the SeqIOA laboratory which is restricted to registered and accredited users, was used to investigate inheritance of variants and visualize bam files. Custom scripts were used on the Auragen. Data from additional cohorts were collected though national collaborative networks in France, or established collaborations with the Broad Institute, the Mefford lab, the Gleeson lab and the EPICARE ERN. We additionally used data from the publicly available gnomAD v3.1.2 (https://gnomad.broadinstitute.org/), and All of Us (https://databrowser.researchallofus.org/) databases. |
| Data analysis | The following software and analysis tool were used: Geneious Prime® 2019.2.3, Seqscape v2.6, STAR aligner v.2.7.11a, CIBERSORTx v1.0, rMATS v.4.3.0, rmats2sashimi, statsmodels v.0.14.5, seaborn v0.13.2, Meffil R package, ggplot2 v3.3.6, pheatmap v1.0.12, stats v4.2.0, and PyMol v3.0.0. Variants were reviewed using MobiDetails (https://mobidetails.iurc.montp.inserm.fr/MD/). Bam files were visualized with IGV 2.19.7. We also used data from Ensembl Release 112 and data from ENCODE Consortium: bigwig files with the plus/minus strand signals of unique reads from the default anisogenic replicate from the following tissues: diencephalon (https://www.encodeproject.org/experiments/ENCSR000AFR/), parietal lobe (https://www.encodeproject.org/experiments/ENCSR000AFY/), occipital lobe, (https://www.encodeproject.org/experiments/ENCSR000AFX/), frontal cortex (https://www.encodeproject.org/experiments/ENCSR000AFS/), temporal lobe (https://www.encodeproject.org/experiments/ENCSR000AGD/), cerebellum (https://www.encodeproject.org/experiments/ENCSR000AFQ/). |

For manuscripts utilizing custom algorithms or software that are central to the research but not yet described in published literature, software must be made available to editors and reviewers. We strongly encourage code deposition in a community repository (e.g. GitHub). See the Nature Portfolio guidelines for submitting code & software for further information.

# Data

Policy information about availability of data

All manuscripts must include a data availability statement. This statement should provide the following information, where applicable:
   - Accession codes, unique identifiers, or web links for publicly available datasets
   - A description of any restrictions on data availability
   - For clinical datasets or third party data, please ensure that the statement adheres to our policy

Variant details have been submitted to ClinVar (SUB15855213; https://www.ncbi.nlm.nih.gov/clinvar/?term=SUB15855213). RNA-seq and methylation data have been deposited in the European Genome-phenome Archive (EGA, http://www.ebi.ac.uk/ega), hosted by the EBI. RNAseq data are available under the Study Accession Number EGAS50000001410 (https://ega-archive.org/studies/EGAS50000001410). Methylation data are accessible under the study accession EGAS00001008070 (https://ega-archive.org/studies/EGAS00001008070). Both are subject to a Data Processing Agreement, and access requests will be reviewed by a Data Access Committee to ensure compliance with ethical and legal standards. Due to ethical considerations, individual genome data cannot be made publicly available. Controlled access is required to safeguard participant privacy and to comply with data protection regulations, including the GDPR in Europe. Access to genome data from the PFMG2025 cohort is governed by French data protection laws and is only possible via the Collecteur Analyseur de Données (CAD). More details can be found on the PFMG2025 website: https://pfmg2025.fr/le-plan/collecteur-analyseur-de-donnees-cad/. The coordinates of the ENCODE Registry of candidate cis-Regulatory Elements (cCREs) in the human genome26 and bigwig files concerning the peaks of histone H3 acetylation of lysine 27 (H3K27Ac) obtained for the H1 human embryonic stem cell line (H1-hESC)27 were downloaded through the UCSC Table Browser60. Bigwig files concerning small RNA-seq data from six human embryonic brain regions were downloaded from the ENCODE portal61 (https://www.encodeproject.org/) with the following identifiers: ENCFF013RLG, ENCFF029RIV, ENCFF034QAV, ENCFF197SSE, ENCFF221KEN, ENCFF343ZBS, ENCFF405QIN, ENCFF532SOY, ENCFF654ONK, ENCFF738LDD, ENCFF870FMA, ENCFF887TOS, ENCFF106ESQ, ENCFF222WBQ, ENCFF250WEA, ENCFF254UEQ, ENCFF319GRF, ENCFF425WUZ, ENCFF443ONL, ENCFF820JTT, ENCFF897IWP, ENCFF915WAC, ENCFF946YVE and ENCFF965GHD.

# Research involving human participants, their data, or biological material

Policy information about studies with human participants or human data. See also policy information about sex, gender (identity/presentation), and sexual orientation and race, ethnicity and racism.

| | |
|---|---|
| Reporting on sex and gender | We obtained information on patients' sex from clinical records and referring physicians. Data on gender were not collected. |
| Reporting on race, ethnicity, or other socially relevant groupings | We collected information on the geographic origin and reported ancestry of a subset of participants; however, these data were not used in any analyses. |
| Population characteristics | Our study report RNU2-2 variants in 141 patients from 122 unrelated families (88 from PFMG cohort and 34 from other cohorts). We collected detailed clinical data for 112 of them ( 55 females and 57 males). The median age at inclusion in the study was 13 years (range: 0 (fetus) - 46 years). |
| Recruitment | The main cohort is composed of patients with rare disorders and their parents when available, who underwent genome sequencing as part of the diagnostic process in France (Plan France Médecine Génomique 2025, PFMG2025).At the time of the study, this cohort comprised short-read genome data from 34,329  patients with rare disorders Analysis of 200 snRNA genes lead to include 42 participants with rare de novo variants in one of these genes and 60 individuals from 52 families with biallelic variants in RNU2-2. Furthermore,  date of 34 additional patients from other cohorts (13 with de novo variants and 21 with biallelic variants) were collected through international collaborations. |
| Ethics oversight | This study was conducted in accordance with the ethical standards and regulations of all participating countries. Written informed consent was obtained for all patients from their parents or legal guardians, with an additional consent form for families agreeing to the publication of photographs. For genetic analyses, patient samples were pseudonymized at each participating center. Information on the patients' sex (but not gender) was extracted from clinical records. The promoters of this research study are Assistance Publique–Hôpitaux de Paris (AP-HP) for hospitals associated with the SeqOIA laboratory (project ID APHP241333) and Grenoble-Alpes University Hospital (CHU Grenoble-Alpes, research ID 19814188) for hospitals affiliated with the Auragen laboratory. Ethical approval was obtained from the University Hospital Essen (24-12010-BO) and the Comité Éthique et Scientifique pour les Recherches, les Études et les Évaluations dans le domaine de la Santé (CESREES; reference 21082803 Bis / 2038764). AP-HP has received an authorization from the Commission Nationale de l'Informatique et des Libertés (CNIL; reference HGTHGT/MFIMFI/AR2426865; request no. 924924336666) for data processing. Additional approvals were obtained from the ethics committee of CHU de Nantes (CCTIRS number 14.556) and from CPP Ouest V (File 06/15, Ref MESR DC 2017 2987; approval date 04/08/2015). For methylation analyses, DNA from patients and controls had been previously collected in a medical context for genetic testing, with written consent including authorization for research use of leftover material. Control samples consisted of individuals without neurodevelopmental disorders, either unaffected relatives or persons tested presymptomatically for other conditions who were found not to carry pathogenic variants. DNA samples used for methylation profiling were stored within the genetics biobank of the CRBi, Rouen, France (collection DC 2008-711, authorization MCRBi/2024/02). The use of these samples was approved by the CERDE ethics committee of Rouen University Hospital (notification E2023-13). Researchers and clinicians from all contributing centers participated throughout the study, from design and implementation to data collection, analysis and manuscript preparation, and are listed as coauthors of this article. |

Note that full information on the approval of the study protocol must also be provided in the manuscript.

# Field-specific reporting

Please select the one below that is the best fit for your research. If you are not sure, read the appropriate sections before making your selection.

☒ Life sciences ☐ Behavioural & social sciences ☐ Ecological, evolutionary & environmental sciences

For a reference copy of the document with all sections, see nature.com/documents/nr-reporting-summary-flat.pdf

# Life sciences study design

All studies must disclose on these points even when the disclosure is negative.

| | |
|---|---|
| Sample size | The main cohort (PFMG) comprised 34,329 patients with rare disorders, including 22,775 individuals with NDD (and their parents when available). These numbers represent all genomes included in the PFMG2025 cohort at the time of study closure (May 2025 for Auragen and November 2025 for SeqOIA). No formal sample size calculation was performed, as this was an exploratory study not designed to test a predefined hypothesis. The same rationale applied to RNA-seq and DNA methylation analyses: biological material (blood samples and extracted DNA) was collected from all families who consented to participate or for whom suitable material was already available. |
| Data exclusions | BAM files were visualized in IGV, and only confirmed de novo variants were included in further analysis. Variants resulting from mismapped reads (regions with multiple variants) or 'de novo' variants also detectable in reads of the parents after inspection of Bam files were discarded. |
| Replication | We replicated the main findings of this study by collecting data from 34 additional patients with de novo or biallelic variants in RNU2-2 from independent cohorts. Moreover, the key results have been independently reproduced by two separate groups (Jackson et al., medRxiv; Greene et al., medRxiv). No findings reported in this study failed independent replication. |
| Randomization | Not applicable (observational study) |
| Blinding | Not applicable (observational study) |

# Reporting for specific materials, systems and methods

We require information from authors about some types of materials, experimental systems and methods used in many studies. Here, indicate whether each material, system or method listed is relevant to your study. If you are not sure if a list item applies to your research, read the appropriate section before selecting a response.

### Materials & experimental systems

| n/a | Involved in the study |
|---|---|
| ☒ | ☐ Antibodies |
| ☒ | ☐ Eukaryotic cell lines |
| ☒ | ☐ Palaeontology and archaeology |
| ☒ | ☐ Animals and other organisms |
| ☒ | ☐ Clinical data |
| ☒ | ☐ Dual use research of concern |
| ☒ | ☐ Plants |

### Methods

| n/a | Involved in the study |
|---|---|
| ☒ | ☐ ChIP-seq |
| ☒ | ☐ Flow cytometry |
| ☒ | ☐ MRI-based neuroimaging |

## Plants

| | |
|---|---|
| Seed stocks | na |
| Novel plant genotypes | na |
| Authentication | na |

