## [Peer Review File · Nature Genetics]

Systematic analysis of snRNA genes reveals frequent RNU2-2 variants in dominant and recessive developmental and epileptic encephalopathies

Corresponding Author: Professor Christel Depienne

Version 0:

Decision Letter:

14th October 2025

Dear Christel,

Your Article "Systematic analysis of snRNA genes reveals frequent RNU2-2 variants in dominant and recessive developmental and epileptic encephalopathies" has been seen by two referees. You will see from their comments below that, while they find your work of interest, they have raised several relevant points. We are interested in the possibility of publishing your study in Nature Genetics, but we would like to consider your response to these points in the form of a revised manuscript before we make a final decision on publication.

To guide the scope of the revisions, the editors discuss the referee reports in detail within the team, including with the chief editor, with a view to identifying key priorities that should be addressed in revision, and sometimes overruling referee requests that are deemed beyond the scope of the current study. In this case, we ask that you clarify specific aspects of the presentation and extend the analyses and discussion as requested, in particular placing the findings in context with the related preprints on bioRxiv. We also invite you to retain the snRNA expression data, at your discretion, as we think these data provide relevant context for the genetic findings. We hope you will find this prioritized set of referee points to be useful when revising your study. Please do not hesitate to get in touch if you would like to discuss these issues further.

We therefore invite you to revise your manuscript taking into account all reviewer and editor comments. Please highlight all changes in the manuscript text file. At this stage, we will need you to upload a copy of the manuscript in MS Word .docx or similar editable format.

*2) If you have not done so already, please begin to revise your manuscript so that it conforms to our Article format instructions, available

[here](http://www.nature.com/ng/authors/article_types/index.html).

*3) Include a revised version of your Reporting Summary: <https://www.nature.com/documents/nr-reporting-summary.pdf>. It will be available to referees (and, potentially, statisticians) to aid in their evaluation if the manuscript goes back for peer review.

Please be aware of our [guidelines](https://www.nature.com/nature-research/editorial-policies/image-integrity) on digital image standards.

EXTENDED DATA FIGURES

Link Redacted

We hope to receive your revised manuscript within 4-8 weeks. If you cannot send it within this time, please let us know.

Nature Genetics is committed to improving transparency in authorship. As part of our efforts in this direction, we are now requesting that all authors identified as 'corresponding author' on published papers create and link their Open Researcher and Contributor Identifier (ORCID) with their account on the Manuscript Tracking System (MTS), prior to acceptance. ORCID helps the scientific community achieve unambiguous attribution of all scholarly contributions. You can create and link your ORCID from the home page of the MTS by clicking on 'Modify my Springer Nature account'. For more information, please visit www.springernature.com/orcid.

Sincerely,
Kyle

Kyle Vogan, PhD
Senior Editor
Nature Genetics
<https://orcid.org/0000-0001-9565-9665>

Referee expertise:

Referee #1: Genetics, neurodevelopmental disorders

Referee #2: Genetics, neurodevelopmental disorders

Reviewers' Comments:

Reviewer #1 (Remarks to the Author):

* Summary of the key results

Description of a recessive RNU2-2 NDD syndrome, some cases of which are due to one de novo variant in one allele and one inherited variant in the other trans allele. The resulting phenotype does not substantially differ from that of the dominant RNU2-2 syndrome. The transcriptome and methylome in blood was not substantially abnormal in the recessive cases. The authors conclude that the combined dominant and recessive RNA2-2-related NDD is as frequent as the dominant RNU4-2-related NDD.

* Originality and significance: if not novel, please include reference

The MS reports a new recessive mendelian disorder due to variants of snRNA. The other 2 groups that have described the same disorder in preprints should be referenced.

medRxiv 2025.09.02.25334957; doi: <https://doi.org/10.1101/2025.09.02.25334957>

medRxiv 2025.08.26.25334179; doi: <https://doi.org/10.1101/2025.08.26.25334179>

* Data & methodology: validity of approach, quality of data, quality of presentation
Detailed and high quality.

* Appropriate use of statistics and treatment of uncertainties

* Conclusions: robustness, validity, reliability

The conclusions are supported by the data. However, the term "gradient-of-impact" that refers to the phenotypic continuum between the dominant and recessive RNU2-2 NDD may not be the best choice. The simple "phenotypic continuum" or "phenotypic overlap" seems simpler and accurate to me.

* Suggested improvements: experiments, data for possible revision.

MAJOR: The study as written contains two different substudies: that of the expression of all snRNAs and the RNU2-2 study. The first topic confuses the reader and it is beyond the important message of the title and abstract. I strongly suggest to delete the section "Identification of potentially functional spliceosomal snRNA genes" L236-L360 and modify the text to deal only with RNU2-2 and RNU4-2. I suggest to publish the expression of all snRNAs in another paper with some additional cell lines and compare the data to those of Mabin, J.W., Lewis, P.W., Brow, D.A. & Dvinge, H. Human spliceosomal snRNA sequence variants generate variant spliceosomes. *RNA* 27, 1186-1203 (2021). In addition, I suggest to eliminate Figure 1 and modify Figure 2 to only contain genes RNU4-2 and RNU2-2.

L474 Please explain the 63/18,000 number (even if it is a repeat of the text in the results). Have you excluded some patients with the de novo variants of L390-312? and are the 18,000 patients all unsolved before or is this the total cohort?

Have the authors studied the clinical presentation of the parents of affected individuals with biallelic RNU2-2 variants in which one allele was de novo or n4G>A?

Please discuss why the snRNA genes may have an increased mutation rate. Could you estimate the mutation rate from your data in nuclear families?

Regarding the phenotypic heterogeneity, could a copy number variation of the RNU2-1 genes be a modifier?

Please discuss the discrepancy between the size of the gene and the frequency of the resulting phenotype.

Comparison of this work with that of the other 2 groups (similarities/difference) needs to be included in the discussion.

* References: appropriate credit to previous work?

Please include the 2 preprints mentioned above.

* Clarity and context: lucidity of abstract/summary, appropriateness of abstract, introduction and conclusions

OK; however, the filtering of variants/cases is complicated and could be clarified by an additional supplementary figure.

Reviewer #2 (Remarks to the Author):

A. Summary of the key results: good

B. Originality and significance: if not novel, please include reference: satisfied

C. Data & methodology: validity of approach, quality of data, quality of presentation: good

D. Appropriate use of statistics and treatment of uncertainties: good

E. Conclusions: robustness, validity, reliability: satisfied

F. Suggested improvements: experiments, data for possible revision: good

G. References: appropriate credit to previous work?: satisfied

H. Clarity and context: lucidity of abstract/summary, appropriateness of abstract, introduction and conclusions: good

This report presents an intriguing study on a recessive neurodevelopmental disorder associated with the RNU2-2 gene, caused by biallelic rare variants. The authors reveal that rare biallelic variants in RNU2-2 are enriched and more frequently transmitted in a cohort of individuals with unresolved neurodevelopmental disorders (NDDs). They achieve this by comparing genome sequencing data from these individuals with large datasets, which helps prioritize candidate disease-causing variants and illustrates the unique genetic architecture of the disorder.

The paper is well-written, and this reviewer has no serious criticisms.

1. Is it possible to classify the severity of the disorder by combining biallelic variant combinations with clinical symptoms? Could there be lethal combinations among these variants?

2. Were there any medications that proved particularly effective in treating epileptic seizures associated with this disorder?

3. Is it feasible to link the genes most affected by gene expression changes to the mechanisms of action of the most effective therapeutic agents?

Version 1:

Decision Letter:

Our ref: NG-A70210R

17th December 2025

Dear Christel,

Your revised manuscript "Systematic analysis of snRNA genes reveals frequent RNU2-2 variants in dominant and recessive developmental and epileptic encephalopathies" (NG-A70210R) has been seen by the original referees. As you will see from the comments below (we note that Reviewer #1 provided only positive Remarks to the Editor at this round of review), they are satisfied with the revision and they have no additional requests, and therefore we will be happy in principle to publish your study in Nature Genetics as an Article pending final revisions to comply with our editorial and formatting guidelines.

We are now performing detailed checks on your paper, and we will send you a checklist detailing our editorial and formatting requirements soon. Please do not upload the final materials or make any revisions until you receive this additional information from us.

Thank you again for your interest in Nature Genetics. Please do not hesitate to contact me if you have any questions.

Sincerely,
Kyle

Kyle Vogan, PhD
Senior Editor
Nature Genetics
<https://orcid.org/0000-0001-9565-9665>

Reviewer #2 (Remarks to the Author):

The authors revised manuscript appropriately. I have no serious criticism.

Version 2:

Decision Letter:

In reply please quote: NG-A70210R1 Depienne

12th February 2026

Dear Christel,

I am delighted to say that your manuscript "Systematic analysis of snRNA genes reveals frequent RNU2-2 variants in dominant and recessive developmental and epileptic encephalopathies" has been accepted for publication in an upcoming issue of Nature Genetics.

Over the next few weeks, your paper will be copyedited to ensure that it conforms to Nature Genetics style. Once your paper is typeset, you will receive an email with a link to choose the appropriate publishing options for your paper, and our Author Services team will be in touch regarding any additional information that may be required.

Your paper will be published online after we receive your corrections and will appear in print in the next available issue. You can find out your date of online publication by contacting the Nature Press Office (press@nature.com) after sending your e-proof corrections.

Authors may need to take specific actions to achieve compliance with funder and institutional open access mandates. If your research is supported by a funder that requires immediate open access (e.g., according to [Plan S principles](https://www.springernature.com/gp/open-science/plan-s-compliance) or the [NIH public access policy](https://www.springernature.com/gp/open-science/us-federal-agency-compliance)), then you should select the gold OA route, and we will direct you to the compliant route where possible. Because authors warrant under our subscription licensing terms that they haven't committed to licensing any version of their article under a license inconsistent with the terms of our agreement – including the applicable embargo period – publication under the subscription model isn't suitable for authors whose funders require no embargo.

If you have not already done so, we strongly recommend that you upload the step-by-step protocols used in this manuscript to protocols.io. protocols.io is an open online resource that allows researchers to share their detailed experimental know-how. All uploaded protocols are made freely available and are assigned DOIs for ease of citation. Protocols can be linked to any publications in which they are used and will be linked to from your article. You can also establish a dedicated workspace to collect all your lab Protocols. By uploading your Protocols to protocols.io, you are enabling researchers to more readily reproduce or adapt the methodology you use, as well as increasing the visibility of your protocols and papers. Upload your Protocols at <https://protocols.io>. Further information can be found at <https://www.protocols.io/help/publish-articles>.

Sincerely,
Kyle

Kyle Vogan, PhD
Senior Editor
Nature Genetics
<https://orcid.org/0000-0001-9565-9665>

Click here if you would like to recommend Nature Genetics to your librarian
<http://www.nature.com/subscriptions/recommend.html#forms>

**Visit the Springer Nature Editorial and Publishing website at http://editorial-jobs.springernature.com?utm_source=ejP_NGen_email&utm_medium=ejP_NGen_email&utm_campaign=ejp_NGen for more information about our career opportunities. If you have any questions, please click [here](mailto:editorial.publishing.jobs@springernature.com).

We thank the two reviewers for their positive evaluation of our manuscript and for providing valuable comments that have helped improve the revised version.

Reviewer #1:

** Summary of the key results*

Description of a recessive RNU2-2 NDD syndrome, some cases of which are due to one de novo variant in one allele and one inherited variant in the other trans allele. The resulting phenotype does not substantially differ from that of the dominant RNU2-2 syndrome. The transcriptome and methylome in blood was not substantially abnormal in the recessive cases. The authors conclude that the combined dominant and recessive RNA2-2-related NDD is as frequent as the dominant RNU4-2-related NDD.

** Originality and significance: if not novel, please include reference*

The MS reports a new recessive mendelian disorder due to variants of snRNA. The other 2 groups that have described the same disorder in preprints should be referenced.

medRxiv 2025.09.02.25334957; doi: <https://doi.org/10.1101/2025.09.02.25334957>

medRxiv 2025.08.26.25334179; doi: <https://doi.org/10.1101/2025.08.26.25334179>

We have now cited and discussed the findings of the two other preprints that were published on MedRxiv around the same time as ours (pages 14-15).

** Data & methodology: validity of approach, quality of data, quality of presentation
Detailed and high quality.*

** Appropriate use of statistics and treatment of uncertainties*

** Conclusions: robustness, validity, reliability*

The conclusions are supported by the data. However, the term "gradient-of-impact" that refers to the phenotypic continuum between the dominant and recessive RNU2-2 NDD may not be the best choice. The simple "phenotypic continuum" or "phenotypic overlap" seems simpler and accurate to me.

The term "gradient-of-impact" does not refer to the phenotypic continuum between dominant and recessive RNU2-2 NDDs but rather to the predicted effects of the variants reported in this study. To clarify, we propose that RNU2-2 variants can be classified into more than just dominant and recessive categories, based on their specific impact on spliceosome function. We predict that some variants will disrupt the maturation cycle of mutant U2-2 RNA, preventing its incorporation into the spliceosome, while others will be incorporated and modify spliceosome function. Variants such as n.4G>A and n.35A>G are predicted to be incorporated into functional spliceosome complexes, acting through a dominant mechanism. Other variants near these positions (e.g., n.5C>A/T, n.20G>A, n.28C>G, n.32T>G...) are also predicted to enter the spliceosome. However, these variants alone do not seem to reach the threshold needed to cause disease with complete penetrance in most cases. Disease manifestation occurs when these variants are paired with another variant in trans or under conditions not yet fully understood, possibly depending on the ratio of other snRNAs present in the spliceosome. On the other end of the spectrum, variants that disrupt the biogenesis of mutant U2-2 RNA fail to mature properly and are not incorporated into the spliceosome. These variants are typically recessive. The combinations of variants observed in recessive cases thus align with a "gradient-of-impact" model, where:

- a) One variant is dominant, being incorporated in the spliceosome and sufficient to cause disease (one non-null allele),
- b) One variant enters the spliceosome but is insufficient alone to cause disease, only doing so when the other allele is non-functional (one non-null allele, one null allele), and possibly other unclear situation,
- c) Variants on both alleles enter the spliceosome and act together (2 non-null alleles),
- d) Variants on both alleles fail to enter the spliceosome (2 null alleles).

We have now revised the text to better reflect this hypothesis, while retaining the term "gradient-of-impact" in the updated version (pages 15-16).

** Suggested improvements: experiments, data for possible revision.*

MAJOR: The study as written contains two different substudies: that of the expression of all snRNAs and the RNU2-2 study. The first topic confuses the reader and it is beyond the important message of the title and abstract. I strongly suggest to delete the section "Identification of potentially functional spliceosomal snRNA genes" L236-L360 and modify the text to deal only with RNU2-2 and RNU4-2. I suggest to publish the expression of all snRNAs in another paper with some additional cell lines and compare the data to those of Mabin, J.W., Lewis, P.W., Brow, D.A. & Dvinge, H. Human spliceosomal snRNA sequence variants generate variant spliceosomes. RNA 27, 1186-1203 (2021). In addition, I suggest to eliminate Figure 1 and modify Figure 2 to only contain genes RNU4-2 and RNU2-2.

Our study does not consist of two separate subparts. The first part, which investigates snRNA functionality and expression, is the foundation for the discoveries made in the second part, as the genetic study focuses on variants in potentially functional snRNA genes within the PFMG cohort. Removing the first part would undermine the rationale and the process by which we arrived at the discovery of the *RNU2-2* dominant and recessive disorders.

Please also note that our approach, which is entirely computational, differs significantly from the method used by Mabin et al. (qPCR) in human cell lines. We believe that the expression of snRNAs requires further study across multiple human tissues, as GTEx data, which is based on polyA+ RNA, is unsuitable for assessing snRNA expression (since most snRNAs are not polyadenylated). Currently, there is no database that accurately captures the expression of snRNA genes. A particular challenge consists in assessing the expression of genes located in different genomic regions that express (almost) identical transcripts (U1, U6...).

To address the reviewer's comment, we have rephrased sections of the manuscript to clarify the logic and rationale behind the study, making the connection between the two parts more explicit (see revised paragraphs pages 8-10). Please note that, due to word limits, some of previous related text has been moved to the Supplementary Notes.

L474 Please explain the 63/18,000 number (even if it is a repeat of the text in the results). Have you excluded some patients with the de novo variants of L390-312? and are the 18,000 patients all unsolved before or is this the total cohort?

We have updated this part of the text and included new figures and tables to clarify the numbers and make the study process more transparent (Extended Data Fig. 1, Supplementary Fig. 4; Supplementary Table 3).

Have the authors studied the clinical presentation of the parents of affected individuals with biallelic RNU2-2 variants in which one allele was de novo or n4G>A?

We have now added a sentence (page 11) indicating cases where a parent who is a heterozygous carrier of a recessive variant presented with milder symptoms. These symptoms may be compatible with reduced penetrance or milder clinical presentations associated with certain variants.

Please discuss why the snRNA genes may have an increased mutation rate. Could you estimate the mutation rate from your data in nuclear families?

We have now updated Supplementary Table 1 to include the 'number of mutations observed / number of mutations expected' ratio, where this data was available (Seplyarskiy et al., *Nature Genetics*, 2023). At this stage, we can only hypothesize about the increased mutation rate observed in snRNA genes. Our current hypothesis is that this elevated mutation rate may be linked to the active, high-level transcription of these genes, which could make them more susceptible to DNA damage and subsequent DNA repair processes. Transcription could generate DSBs, and the subsequent repair may involve nearby paralogous sequences, which could lead to localized hypermutability. This supports a model where hypermutability is not specific to RNUs but instead reflects resolution of transcription-associated damage through non-homologous repair processes. However, this hypothesis remains to be experimentally validated.

Regarding the phenotypic heterogeneity, could a copy number variation of the RNU2-1 genes be a modifier?

We thank the reviewer for raising this interesting hypothesis. However, assessing whether short-read data can reliably determine the copy number of U2-1 genes on each allele would require an entirely new study. To accurately determine U2 copy number on each allele, long-read technologies should be used in parallel. Furthermore, it remains unclear whether an increased U2-1 gene copy number is associated with U2-1 higher expression, which would be crucial for understanding how U2-1 copy number might influence RNU2-2-associated NDD.

Another possibility is that variants in some RNU2-1 genes comprised in the satellite, which are currently inaccessible to analysis in short-read genomes, could modify the penetrance or expressivity of RNU2-2 variants. However, short-read data are also not suitable for investigating this hypothesis, as reads mapping to RNU2-1 genes typically have a mapping quality score of zero or are discarded, depending on the bioinformatic pipeline used for alignment to the reference genome. Therefore, investigating both hypotheses would require long-read sequencing technologies on a significant number of RNU2-2 patients, and thus falls outside the scope of this study.

Please discuss the discrepancy between the size of the gene and the frequency of the resulting phenotype.

The discrepancy between the size of snRNA genes and the frequency of the resulting phenotypes is possibly linked to hypermutability of snRNA compared to protein-coding genes. We have now included a sentence discussing this aspect page 14.

Comparison of this work with that of the other 2 groups (similarities/difference) needs to be included in the discussion.

We have added several paragraphs in the discussion comparing our findings to those of the two other preprints (pages 14-16).

** References: appropriate credit to previous work?
Please include the 2 preprints mentioned above.*

Response: We have now included the two preprints as references.

** Clarity and context: lucidity of abstract/summary, appropriateness of abstract, introduction and conclusions
OK; however, the filtering of variants/cases is complicated and could be clarified by an additional supplementary figure.*

We have included an additional figure (Extended Data Fig. 1) to clarify the study design and the variant filtering process.

Reviewer #2:

- A. Summary of the key results: good*
- B. Originality and significance: if not novel, please include reference: satisfied*
- C. Data & methodology: validity of approach, quality of data, quality of presentation: good*
- D. Appropriate use of statistics and treatment of uncertainties: good*
- E. Conclusions: robustness, validity, reliability: satisfied*
- F. Suggested improvements: experiments, data for possible revision: good*
- G. References: appropriate credit to previous work?: satisfied*
- H. Clarity and context: lucidity of abstract/summary, appropriateness of abstract, introduction and conclusions: good*

This report presents an intriguing study on a recessive neurodevelopmental disorder associated with the RNU2-2 gene, caused by biallelic rare variants. The authors reveal that rare biallelic variants in RNU2-2 are enriched and more frequently transmitted in a cohort of individuals with unresolved neurodevelopmental disorders (NDDs). They achieve this by comparing genome sequencing data from these individuals with large datasets, which helps prioritize candidate disease-causing variants and illustrates the unique genetic architecture of the disorder.*

The paper is well-written, and this reviewer has no serious criticisms.

1. Is it possible to classify the severity of the disorder by combining biallelic variant combinations with clinical symptoms? Could there be lethal combinations among these variants?

Response: We have tried different approaches to investigate genotype-phenotype correlations (see revised Extended Data Fig. 7c-d) but were unable to identify any significant associations with the location of the variants, their combinations, their status (compound heterozygous or homozygous), or their predicted impact. While we observe that some biallelic associations are linked to more severe phenotypes, the clinical variability within each group seems to exceed and mask the individual effect of the variants. We believe that the true impact of these variants may also depend on additional factors (e.g., variants in other snRNA genes).

2. Were there any medications that proved particularly effective in treating epileptic seizures

associated with this disorder?

Response: Patients were treated based on seizure type, as is standard clinical practice, rather than on the knowledge of the *RNU2-2* variants, which only became available recently. We have now added a sentence to address this point in the results (clinical section). Given the clinical variability discussed earlier, it is currently challenging to determine which drug is most effective. This aspect clearly warrants further investigation in a dedicated study.

3. Is it feasible to link the genes most affected by gene expression changes to the mechanisms of action of the most effective therapeutic agents?

Response: The RNA-seq study was conducted in activated lymphocytes, a cell type that is not ideal for addressing this question, as anti-epileptic drugs typically target ion channels or pathways that are neuron- or brain-specific. However, the revised version now includes the list of genes with splicing anomalies detected by RNAseq in Supplementary Tables 8-10. These tables provide readers with a set of genes of interest that they may find relevant for exploring potential links to drug efficacy.